# The co-occurrence of genetic variants in the *TYR* and *OCA2* genes confers susceptibility to albinism

David J. Green[1], Vincent Michaud [2,3], Eulalie Lasseaux[2], Claudio Plaisant[2], UK Biobank Eye and Vision Consortium*, Tomas Fitzgerald [4], Ewan Birney [4], Graeme C. Black[1,5,7], Benoît Arveiler [2,3,7] & Panagiotis I. Sergouniotis [1,4,5,6] ✉

Although rare genetic conditions are mostly caused by DNA sequence alterations that functionally disrupt individual genes, large-scale studies using genome sequencing have started to unmask additional complexity. Understanding how combinations of variants in different genes shape human phenotypes is expected to provide important insights into the clinical and genetic heterogeneity of rare disorders. Here, we use albinism, an archetypal rare condition associated with hypopigmentation, as an exemplar for the study of genetic interactions. We analyse data from the Genomics England 100,000 Genomes Project alongside a cohort of 1120 individuals with albinism, and investigate the effect of dual heterozygosity for the combination of two established albinism-related variants: *TYR*:c.1205 G > A (p.Arg402Gln) [rs1126809] and *OCA2*:c.1327 G > A (p.Val443Ile) [rs74653330]. As each of these changes alone is insufficient to cause disease when present in the heterozygous state, we sought evidence of synergistic effects. We show that, when both variants are present, the probability of receiving a diagnosis of albinism is significantly increased (odds ratio 12.8; 95% confidence interval 6.0 – 24.7; *p*-value 2.1 ×10⁻⁸). Further analyses in an independent cohort, the UK Biobank, support this finding and highlight that heterozygosity for the *TYR*:c.1205 G > A and *OCA2*:c.1327 G > A variant combination is associated with statistically significant alterations in visual acuity and central retinal thickness (traits that are considered albinism endophenotypes). The approach discussed in this report opens up new avenues for the investigation of oligogenic patterns in apparently Mendelian disorders.

An individual affected by a condition that appears to segregate as a Mendelian trait would be expected to carry large-effect genetic variation in a single gene/locus. However, for a significant proportion of affected probands, single-gene-centric genomic analyses fail to identify a clear genetic diagnosis[1–6]. Although this diagnostic and knowledge gap is, to a degree, due to imperfect phenotyping, sequencing or

[1]Division of Evolution, Infection and Genomics, School of Biological Sciences, Faculty of Biology, Medicine and Health, University of Manchester, Manchester, UK. [2]Department of Medical Genetics, University Hospital of Bordeaux, Bordeaux, France. [3]INSERM U1211, Rare Diseases, Genetics and Metabolism, University of Bordeaux, Bordeaux, France. [4]European Molecular Biology Laboratory, European Bioinformatics Institute (EMBL- EBI), Wellcome Genome Campus, Cambridge, UK. [5]Manchester Centre for Genomic Medicine, Saint Mary's Hospital, Manchester University NHS Foundation Trust, Manchester, UK. [6]Manchester Royal Eye Hospital, Manchester University NHS Foundation Trust, Manchester, UK. [7]These authors contributed equally: Graeme C. Black, Benoît Arveiler. *A list of authors and their affiliations appears at the end of the paper. ✉e-mail: panagiotis.sergouniotis@manchester.ac.uk

annotation, other factors may be contributing. There is, for example, growing evidence supporting the hypothesis that many apparently Mendelian phenotypes are caused by the interaction of multiple variants in more than one locus[7–11]. Uncovering such genetic interactions has the potential to provide insights not only into disease mechanisms and diagnostics but also into phenomena like low penetrance and variable expressivity[12–14].

There are many patterns through which genetic changes in more than one gene can synergistically shape a phenotype. A simple scenario would involve two heterozygous variants in two distinct loci cooperating to mediate disease (when each individual variant in isolation fails to explain the clinical presentation). Only a small number of cases with this specific form of digenic inheritance have been described[15–17], the majority detected through qualitative approaches involving variant segregation analyses in a small number of families. In these cases, interaction-based models have generally been used to provide a *post hoc* explanation of the observed phenomena.

In this work, we describe a quantitative case-control approach that allows obtaining statistical evidence of digenic patterns. We focus on specific genotypes in the *TYR* and *OCA2* genes; biallelic variants in each of these two genes are associated with albinism, a clinically and genetically heterogeneous group of conditions characterised by reduced levels of melanin pigment. Key features of albinism include visual abnormalities and under-development of the central retina (*i.e.* the fovea); skin and/or hair hypopigmentation may also be present[18]. Expanding recent work[19–22], we further advance our understanding of the genetic complexity of this rare disorder.

## Results and Discussion

A cohort of 1015 people with albinism underwent testing of ≤19 albinism-related genes; these individuals were not known to be related and had predominantly European-like ancestries (Supplementary Data 1). A further 105 probands with albinism were identified in the Genomics England 100,000 Genomes Project dataset[23]. A "control" cohort of 29,451 unrelated individuals that had no recorded diagnosis or features of albinism was also identified in this resource (Fig. 1 and Supplementary Table 1; "Methods").

Searching for genetic elements underlying digenic traits may be carried out at the level of genes, genotypes or DNA variants. It is noted that a given pair of DNA variants comprises up to nine pairs of genotypes while a given gene may contain a large number of variants. Each of these search strategies has inherent advantages and pitfalls[24]. We opted for a genotype-based approach that offers a high level of precision and is more likely to provide insights that can be used for clinical genome interpretation.

Our genotype-based analysis focused on two prevalent albinism-related changes, *TYR*:c.1205 G > A (p.Arg402Gln) [rs1126809] and *OCA2*:c.1327 G > A (p.Val443Ile) [rs74653330]. We hypothesised that when these two missense variants are both present in the heterozygous state ("dual heterozygosity"), their co-occurrence is driving the pathology observed in albinism. Notably, both these changes have been shown to reduce the activity of their respective protein products in vitro[25–29]. Previous studies have also pointed to a functional interaction between the TYR and OCA2 proteins: the latter has a role in the pH regulation of the melanosome which, in turn, can affect the catalytic activity of the former (*i.e.* tyrosinase, the rate-limiting enzyme for the synthesis of melanin in the melanosomes of retinal pigment epithelia and melanocytes)[18,30]. Both *TYR*:c.1205 G > A and *OCA2*:c.1327 G > A have been implicated in albinism when in trans with pathogenic changes in the same gene[21,31]. Also, multiple associations have been recorded for these two variants including with skin/hair pigmentation and skin cancer[32]. It is noted that these two changes were chosen as they are the two commonest albinism-related variants in populations of European-like ancestries, a feature that can help mitigate potential issues with statistical power. It is also

highlighted that the minor allele frequencies of the *TYR*:c.1205 G > A and *OCA2*:c.1327 G > A changes in European populations are 27% and 0.3%, respectively (Non-Finnish European subset of the Genome Aggregation Database [gnomAD] v2.1.1)[33]. The particularly common *TYR*:c.1205 G > A change has been associated with low penetrance (and variable expressivity), and previous studies have pointed to the potential for other *TYR* variants to modify its effect[19–21,25,34–36].

We used Firth regression analysis[37,38] to study how dual heterozygosity for *TYR*:c.1205 G > A and *OCA2*:c.1327 G > A affects the probability of receiving a diagnosis of albinism ("risk of albinism"). This logistic regression approach has been designed to handle small, imbalanced datasets (which are common in studies of rare conditions) and allows for adjustment of key covariates (which is not possible in contingency table methods) (Fig. 1). To increase the genetic similarity[39] between the compared cohorts (cases and controls) and to reduce the impact of albinism-related variants other than the two studied changes (*TYR*:c.1205 G > A and *OCA2*:c.1327 G > A), we excluded from our analysis individuals that: (i) did not have European-like ancestries, (ii) were found to have genotypes in keeping with a molecular diagnosis of albinism, (iii) carried pathogenic or likely pathogenic variants in the *TYR* or *OCA2* gene (with the exception of *TYR*:c.1205 G > A and *OCA2*:c.1327 G > A). The resulting case and control cohorts included 204 and 20,350 individuals, respectively. These were used to perform case-control comparisons between the following genotype groups:

- *group A* (reference group): homozygous for *TYR*:c.1205 = and *OCA2*:c.1327 =.
- *group B* (single *TYR* heterozygote group): heterozygous for *TYR*:c.1205 G > A and homozygous for *OCA2*:c.1327 =.
- *group C* (single *OCA2* heterozygote group): homozygous for *TYR*:c.1205 G = and heterozygous for *OCA2*:c.1327 G > A.
- *group D* (dual heterozygote group): heterozygous for the *TYR*:c.1205 G > A and *OCA2*:c.1327 G > A variant combination.

These groups correspond to different levels of dysfunction of the TYR and OCA2 molecules, and the findings of the regression analysis revealed a synergistic potentiation of the changes at the two loci (Fig. 2, Supplementary Fig. 1a and Supplementary Table 2). The point estimate for the odds of receiving a diagnosis of albinism in individuals who are heterozygous for the *TYR*:c.1205 G > A and *OCA2*:c.1327 G > A variant combination was 12.8 (the 95% confidence interval was 6.0 – 24.7, and the p-value was $2.1 \times 10^{-8}$).

To increase confidence in these observations, we performed additional analyses in an independent cohort, the UK Biobank[40]. We found that UK Biobank volunteers who were heterozygous for the *TYR*:c.1205 G > A and *OCA2*:c.1327 G > A variant combination had not only a higher chance of receiving a diagnosis of albinism (OR > 4.2, p-value = 0.005) but also had, on average, slightly worse visual acuity and marginally thicker central retina (the respective Kruskal-Wallis p-values were 0.002 and 0.004; Fig. 3 and Supplementary Tables 3–6). It is noted that visual acuity and central retinal thickness are quantitative endophenotypes of albinism and these findings suggest that the studied *TYR*/*OCA2* variant combination is both contributing to the genetic architecture of albinism and having a role in the development of the visual system in individuals from the general population.

We provide evidence that the *TYR*:c.1205 G > A and *OCA2*:c.1327 G > A changes appear to have a joint effect when both are encountered in the heterozygous state. However, the interpretation of this variant combination in clinical diagnostic settings should be approached cautiously. Although the detected effect size (odds ratio) is significant, it would be more in keeping with a predisposing[41] rather than a pathogenic[42] genotype. Further, while it is tempting to view our observations as an example of a digenic pattern, this would have reductionist connotations. Interactions are ubiquitous in nature, and we expect that increased sample sizes will elucidate longer "chains" of interconnected genetic variation. It is thus plausible that the effect size

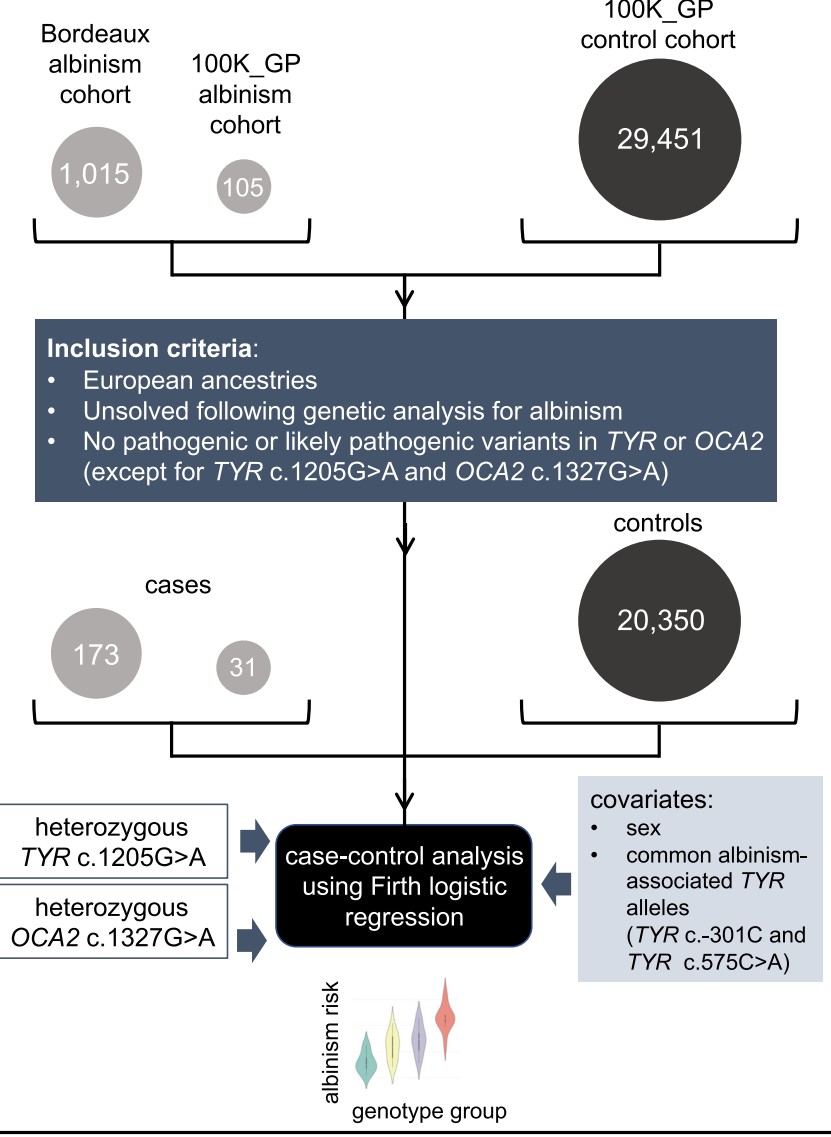

**Fig. 1 | Outline of the case-control study design.** A case-control analysis was performed to gain insights into the role of genotypes involving dual heterozygosity for *TYR*:c.1205 G > A (p.Arg402Gln) [rs1126809] and *OCA2*:c.1327 G > A (p.Val443Ile) [rs74653330] in albinism. These two missense changes were selected as they are the commonest albinism-related variants in European-like populations. The majority of participants in the "case" cohort (1015/1120) were identified through the database of the University Hospital of Bordeaux Molecular Genetics Laboratory, France. All these probands had at least one key ocular feature of albinism, *i.e.* nystagmus or prominent foveal hypoplasia (see "Methods"). The remaining 105/1120 cases were identified through the Genomics England 100,000 Genomes Project dataset and had a diagnosis of albinism or partial/ocular albinism. The "control" cohort included 29,451 unrelated individuals from the Genomics England 100,000 Genomes Project dataset, none of whom had a recorded diagnosis of albinism. To reduce the likelihood of obtaining spurious signals due to population stratification effects or due to the presence of albinism-related variants other than the two studied changes, we focused only on individuals who: (i) were projected to have European-like ancestries; (ii) did not have a genotype in keeping with a molecular diagnosis of albinism; (iii) were not heterozygous for a pathogenic or a likely pathogenic variant in *TYR* or *OCA2* (with the exception of *TYR*:c.1205 G > A and *OCA2*:c.1327 G > A). The following two covariates were used: sex and number of common albinism-associated alleles, *i.e.* DNA sequence alterations in the genomic locations corresponding to *TYR*:c.– 301 C > T [rs4547091] and *TYR*:c.575 C > A (p.Ser192Tyr) [rs1042602]. These two variants have been previously shown to modify the effect of *TYR*:c.1205 G > A, a common missense change that can act as a low penetrance variant[19–21]. To assess the robustness of the findings we performed additional analyses in subsets of the cohort. Validation studies in an independent cohort (UK Biobank) were also conducted. 100K_GP, Genomics England 100,000 Genomes Project; het, heterozygous; UKB, UK Biobank. *TYR* and *OCA2* variant numbering was based on the transcripts with the following identifiers: NM_000372.5/ ENST00000263321.6. and NM_000275.3/ENST00000354638.8.

of the studied genotype will increase when further genetic changes that impact tyrosinase function or melanosome pH are added to the model. Indeed, observations in the presented data already point to this possibility (Supplementary Fig. 3 and Supplementary Data 2).

One limitation of this study is our inability to match stringently the albinism cases with the unaffected controls, especially in terms of genetic background[43]. Although recent ancestry was considered, our analysis was imperfect as it was not possible to reliably assign genetic ancestry to most albinism cases. We used a combination of orthogonal

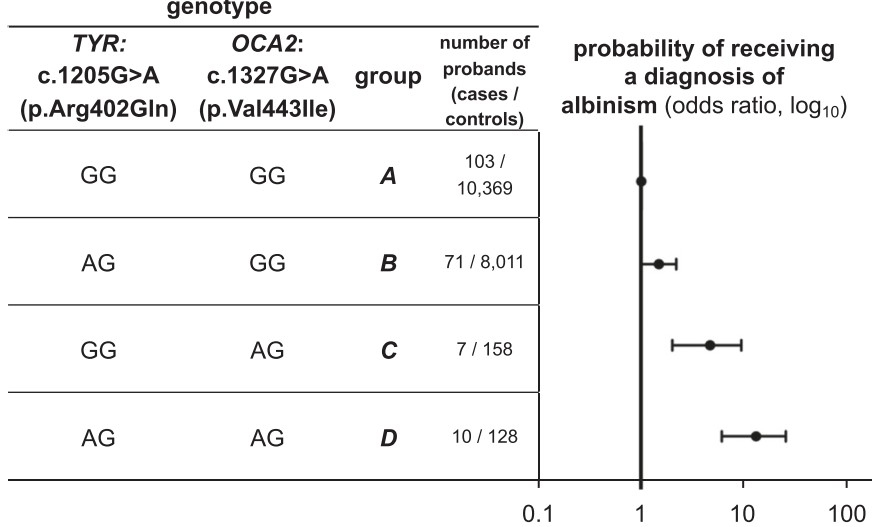

**Fig. 2 | The combination of the *TYR*:c.1205 G > A (p.Arg402Gln) and *OCA2*:c.1327 G > A (p.Val443Ile) variants in a dual heterozygous state confers susceptibility to albinism.** The probability of receiving a diagnosis of albinism (*i.e.* the risk of albinism) was calculated for the following genotype groups: (i) ***group A*** (reference group): homozygous for *TYR*:c.1205 = and *OCA2*:c.1327 =; (ii) ***group B*** (single *TYR* heterozygote group): heterozygous for *TYR*:c.1205 G > A and homozygous for *OCA2*:c.1327 =; (iii) ***group C*** (single *OCA2* heterozygote group): homozygous for *TYR*:c.1205 G = and heterozygous for *OCA2*:c.1327 G > A; (iv) ***group D*** (dual heterozygote group): heterozygous for the *TYR*:c.1205 G > A and *OCA2*:c.1327 G > A variant combination. A log₁₀ scale is used. The circle in the middle of each horizontal line (95% confidence interval) represents the point estimate of each odds ratio. Group A was used as the reference group to which all other groups were compared using Firth regression analysis (*i.e.* the odds ratio for this group was fixed at 1). Further information, including numerical data, can be found in Supplementary Table 2.

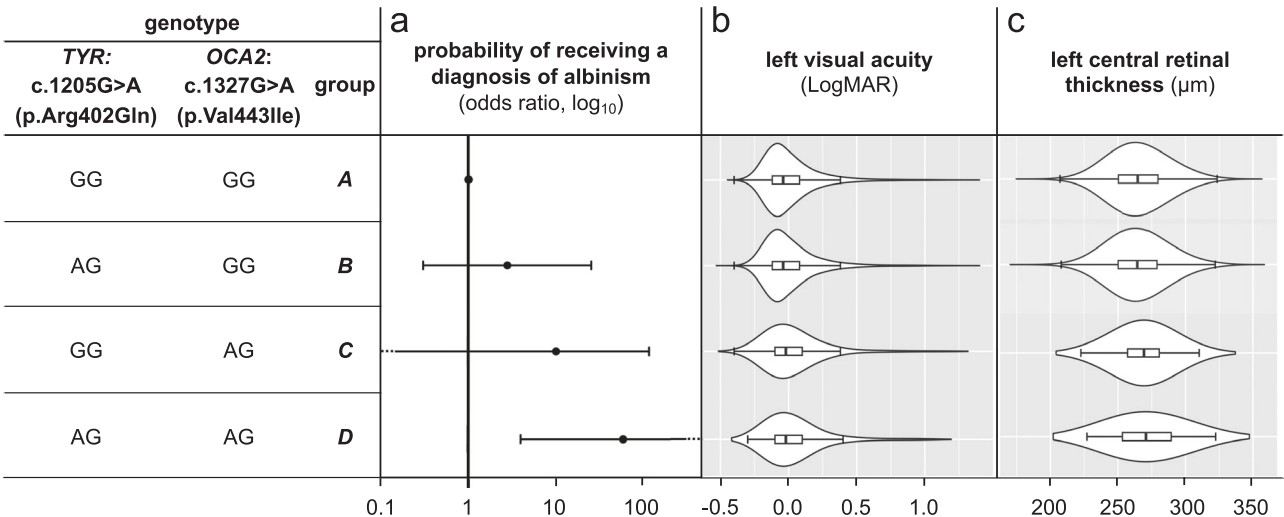

**Fig. 3 | R eplication studies in the UK Biobank cohort.** The combination of the *TYR*:c.1205 G > A (p.Arg402Gln) and *OCA2*:c.1327 G > A (p.Val443Ile) variants in a dual heterozygous state is associated with (**a**) a higher probability of receiving an albinism diagnosis (i.e. a higher risk of albinism), (**b**) lower visual acuity and (**c**) increased central retinal thickness in UK Biobank participants. The following genotype groups were studied: (i) **group A** (reference group): homozygous for *TYR*:c.1205 = and *OCA2*:c.1327 =; (ii) **group B** (single *TYR* heterozygote group): heterozygous for *TYR*:c.1205 G > A and homozygous for *OCA2*:c.1327 =; (iii) **group C** (single *OCA2* heterozygote group): homozygous for *TYR*:c.1205 G = and heterozygous for *OCA2*:c.1327 G > A; (**iv**) **group D** (dual heterozygote group): heterozygous for the *TYR*:c.1205 G > A and *OCA2*:c.1327 G > A variant combination. In (**3a**), a log₁₀ scale is used. The circle in the middle of each horizontal line (95% confidence interval) represents the point estimate of each odds ratio. In (**3b** and **c**), the bounds of each rectangular box correspond to the upper and lower quartile, representing the interquartile range and showing where the central 50% of the data lies. The vertical line inside each box denotes the median value. The edges of the whiskers extending from each box correspond to the most extreme data points (that are not considered outliers). It is noted that individuals with albinism tend to have reduced visual acuity (i.e. higher LogMAR value than 0.2) and increased central retinal thickness (due to underdevelopment of the fovea). More details on the utilised approach can be found in the Methods and in Supplementary Fig. 2. Further information, including numerical data, can be found in Supplementary Tables 3–6.

approaches to evaluate the robustness and generalisability of our findings. First, we used 35 presumed neutral single-nucleotide variants to calculate the genomic inflation factor lambda (λGC)[44,45]; the $\lambda_{median}$ was found to be 1.04, in keeping with limited confounding by ancestry (Supplementary Table 7). Subsequently, we performed a targeted

secondary analysis using only Genomics England 100,000 Genomes Project data which allowed us to study case and control groups of high genetic similarity. The results supported our key findings and increased confidence in the validity of the detected associations (Supplementary Fig. 1b and Supplementary Table 8). Another potential

caveat is around the granularity of the phenotypic information used for case definition ("Methods"; Supplementary Table 10). The availability of imaging/electrophysiological data in the present cohort, for instance, did not match that of certain smaller-scale studies[46,47]. Nonetheless, the prevalence of key albinism-related phenotypic features in the main dataset was comparable to that in other relevant cohorts[46–49] (Supplementary Fig. 4). To further reduce the likelihood of an alternative explanation/diagnosis in the *TYR*:c.1205 G > A and *OCA2*:c.1327 G > A dual heterozygotes (Supplementary Data 3), additional genetic studies were undertaken with these confirming that the relevant individuals do not carry disease-implicated variants in *PAX6* or in genes associated with achromatopsia or congenital stationary night blindness. A third caveat is that the detected signal in dual heterozygotes for *TYR*:c.1205 G > A and *OCA2*:c.1327 G > A could, to a degree, be driven by undetected pathogenic variation in either *TYR* or *OCA2*. This cannot be excluded despite the fact that medical-grade genetic analyses were undertaken. It can be argued, however, that significant confounding is unlikely especially given that the effect of dual heterozygosity for *TYR*:c.1205 G > A and *OCA2*:c.1327 G > A appears to be consistently greater than that of genotypes involving heterozygosity for either *TYR*:c.1205 G > A or *OCA2*:c.1327 G > A alone (Supplementary Tables 2, 3, 8 and 9). Nonetheless, statistically significant results were also obtained for individuals carrying the *OCA2*:c.1327 G > A change in heterozygous state (without *TYR*:c.1205 G > A) and future studies will provide further insights into the extent to which this change alone drives the observed signal.

In conclusion, we have shown that dual heterozygosity for a *TYR* and an *OCA2* variant confers susceptibility to albinism. Shedding light on this and other variant interactions is expected to increase our understanding of the pathways and cellular processes affected by albinism. Notably, the concepts discussed herein are likely to be relevant to other rare disorders, and systems-based approaches using biological modules are expected to narrow the molecular diagnostic gap. Future work will focus on increasing sample sizes, embracing more diverse populations and searching for high-order interactions by considering more than two variants at a time.

## Methods

### Ethics approval
Informed consent was obtained from all participants or their parents in the case of minors. The study of individuals from the University Hospital of Bordeaux albinism cohort has been approved by the relevant local ethics committee (Comité de Protection des Personnes Sud-Ouest et Outre Mer III, Bordeaux, France). The informed consent process for the Genomics England 100,000 Genomes Project has been approved by the National Research Ethics Service Research Ethics Committee for East of England – Cambridge South Research Ethics Committee. The UK Biobank has received approval from the National Information Governance Board for Health and Social Care and the National Health Service North West Centre for Research Ethics Committee (Ref: 11/NW/0382). All investigations were conducted in accordance with the tenets of the Declaration of Helsinki.

### Cohort characteristics and genotyping
**University Hospital of Bordeaux albinism cohort**. Individuals with albinism were identified through the database of the University Hospital of Bordeaux Molecular Genetics Laboratory, France. This is a national reference laboratory that has been performing genetic testing for albinism since 2003 and has been receiving samples from individuals predominantly based in France (or French-administered overseas territories).

Information on the dermatological and ophthalmological phenotypes was available for affected individuals. Among others, age-appropriate visual acuity was recorded, and key ocular features of albinism – such as nystagmus, iris translucency (none; peripheral or

few iris transillumination defects; diffuse iris transillumination defects)[48] and foveal hypoplasia (none; hypoplasia / aplasia)[48,49] – were evaluated. In addition to visual inspection for these features, imaging of the central macula using spectral domain or swept-source optical coherence tomography (OCT) platforms was performed in a subset of study participants. Attempts to obtain OCT scans through the foveal centre were made at all times. Relevant images were not acquired in young children and in cases with suboptimal cooperation. The clinical data were assessed independently by two clinicians with experience in the field of albinism and only cases with a consensus clinical impression of albinism were included (Supplementary Table 10). It is noted that each of these cases had at least one of the key ocular features of albinism, *i.e.* infantile nystagmus or prominent foveal hypoplasia (corresponding to high-grade foveal hypoplasia on OCT ($\geq 2$)[50] and/or to the absence of the foveal depression/reflex as determined by stereoscopic view of the macula by ophthalmoscopy or examination of fundus photographs[51]). No pre-screening based on genotype was undertaken other than selecting individuals in whom the position c.– 301 of *TYR* was sequenced. Only individuals who were not knowingly related were included.

Genetic testing, bioinformatic analyses, and variant interpretation were performed as previously described[20,21]. Briefly, most participants had gene-panel testing of ≤19 genes associated with albinism (*C10ORF11*, *GPR143*, *HPS 1* to 10, *LYST*, *OCA2*, *SLC38A8*, *SLC24A5*, *SLC45A2*, *TYR*, *TYRP1*) using IonTorrent platforms. High-resolution array-CGH (comparative genomic hybridisation) was used to detect copy number variants in these genes. All genetic changes of interest were confirmed with an alternative method (e.g., Sanger sequencing or quantitative PCR). Clinical interpretation of variants was performed using criteria consistent with the 2015 American College of Medical Genetics and Genomics (ACMG) best practice guidelines[42]. It is noted that genetic findings in this cohort have been partly reported in previous publications from our group[19,22] and that comprehensive genomic testing (including single-nucleotide variant and copy number variant analysis of all the aforementioned 19 albinism-related genes) was performed in all cases that remained unsolved.

Due to the limited number of genomic loci screened in this cohort, it was not possible to reliably assess genetic ancestry and to "objectively" assign individuals to genetic ancestry groups. Attempting to mitigate this, we processed available data on self-identified ethnicity that were collected through questionnaires. Responses were inspected, and stratification into five broad continental groups (European, African, Admixed American, East Asian, and South Asian) was performed.

**Genomics England 100,000 Genomes Project cohort**. Clinical and genomic data from the Genomics England 100,000 Genomes Project were accessed through a secure Research Environment that is available to registered users. This dataset was collected as part of a national genome sequencing initiative[52]. Enrolment was coordinated by Genomics England Limited, and participants were recruited mainly at National Health Service (NHS) Hospitals in the UK[23]. Clinical information was recorded in Human Phenotype Ontology (HPO)[53] terms and International Classification of Diseases (ICD) codes. Genome sequencing was performed in DNA samples from ≥ 78,195 individuals using Illumina HiSeq X systems (150 base-pair paired-end format). Reads were aligned using the iSAAC Aligner v03.16.02.19, and small variants were called using Starling v2.4.7[54]. Structural variants and long indel (> 50 bp) calling were performed with Manta v0.28.0, and copy number variants were called with Canvas v1.3.1[55,56]. Aggregation of single-sample gVCFs was performed using the Illumina software gVCF genotyper v2019.02.29; normalisation/decomposition was implemented by vt v0.57721[57]. The multi-sample VCF was then split into 1,371 roughly equal chunks to allow faster processing, and the loci of interest were queried using bcftools v1.9[58] (see https://research-help.genomicsengland.co.uk/display/GERE/ for further

information). Only variants that passed standard quality control criteria were processed (including the criteria discussed in https://re-docs.genomicsengland.co.uk/site_qc/ and criteria analogous to those outlined in https://re-docs.genomicsengland.co.uk/somatic_sv/#note-on-variant-calling). In addition, we filtered out genotypes with: genotype scores < 20; read depth < 10; allele balance < 0.2 and > 0.8 for heterozygotes; allele balance > 0.1 or < 0.9 for homozygotes (reference and alternate, respectively). Genomic annotation was performed using Ensembl VEP[59]. Variants with a minor allele frequency (MAF) < 1% in the Genome Aggregation Database (gnomAD v2.1.1)[33] and a "disease-causing" (DM) label in the Human Gene Mutation Database (HGMD) v2021.2[60] were flagged (and are hitherto referred to as "HGMD-listed variants"). In-depth manual curation of all detected rare (i.e. MAF < 1%) variants in *TYR* and *OCA2* was undertaken, and these changes were classified according to the 2015 ACMG best practice guidelines[42].

Ancestry inference was performed in the Genomics England 100,000 Genomes Project cohort using principal component analysis. Data from the 1000 Genomes Project (phase 3) dataset were used, and five broad super-populations were projected (European, African, Admixed American, East Asian, and South Asian) (further information on this can be found online at https://research-help.genomicsengland.co.uk/display/GERE/Ancestry+inference).

We focused on a pre-determined subset of the Genomics England 100,000 Genomes Project dataset that includes only unrelated probands (n = 29,556). Of these, 105 had a diagnosis of albinism, i.e. the ICD-10 term "Albinism" [E70.3] and/or the HPO terms "Albinism" [HP:0001022], "Partial albinism" [HP:0007443] or "Ocular albinism" [HP:0001107] were assigned (Supplementary Table 10). Together with the University Hospital of Bordeaux cases, these 105 probands formed the "case" cohort. The remaining 29,451 probands had no recorded diagnosis or phenotypic features of albinism and formed the "control" cohort. We note that comprehensive ophthalmic phenotyping was not routinely undertaken in Genomics England 100,000 Genomes Project participants. Therefore, we cannot be certain that a small number of individuals with mild/subclinical forms of albinism are not included in the control cohort.

## Case-control analysis to estimate the probability of an albinism diagnosis

The effect of dual heterozygosity for *TYR*:c.1205 G > A (p.Arg402Gln) [rs1126809] and *OCA2*:c.1327 G > A (p.Val443Ile) [rs74653330] on the probability of receiving a diagnosis of albinism (i.e. the albinism risk) was estimated using data from the University Hospital of Bordeaux albinism cohort and the Genomics England 100,000 Genomes Project dataset. A case-control analysis of a binary trait (presence/absence of albinism) was conducted assuming a recessive model. Logistic regression using the Firth bias reduction method[37,38] was utilised (as implemented in the "logistf" R package). The following parameters were set:

- inclusion criteria: the analysis was conducted only on study subjects who: (i) were projected to have European-like ancestries, (ii) were not found to have genotypes in keeping with a molecular diagnosis of albinism, (iii) carried no pathogenic or likely pathogenic variants in the *TYR* or *OCA2* gene (with the exception of the two studied changes *TYR*:c.1205 G > A and *OCA2*:c.1327 G > A).
- covariates: sex; the number of common albinism-associated alleles (see relevant sub-section below).

**Ancestry.** To increase the genetic similarity between the studied case and control cohorts, we focused on individuals who were likely to fall within the same broad ancestral group. It is highlighted, though, that the utilised approach was imperfect as it was not possible to objectively determine genetic ancestry in individuals from the University

Hospital of Bordeaux albinism cohort (as mentioned above, self-identified ethnicity was instead used as a surrogate). Confounding by ancestry (i.e. population stratification) is, therefore, a possibility[39]. We attempted to quantify this by calculating the genomic inflation factor lambda (λGC). A total of 35 single-nucleotide variants were selected using a previously described approach (see Supplementary Table 7 and[18]). Case-control comparisons were then made for each of these 35 λGC markers using Firth regression analysis. The resulting test statistics were used to calculate the median value of λGC[18].

**Albinism-related genotypes.** To reduce the likelihood of obtaining spurious results due to the impact of albinism-related variants other than the two studied changes (*TYR*:c.1205 G > A and *OCA2*:c.1327 G > A), we excluded individuals who carried genotypes consistent with a molecular diagnosis of albinism. This included study subjects with:

- homozygous pathogenic or likely pathogenic HGMD-listed variants in autosomal albinism-related genes *(TYR, OCA2, TYRP1, SLC45A2, SLC24A5, C10ORF11, HPS 1 to 10, LYST, SLC38A8)*,
- presumed biallelic pathogenic or likely pathogenic HGMD-listed variants in autosomal albinism-related genes *(TYR, OCA2, TYRP1, SLC45A2, SLC24A5, C10ORF11, HPS 1 to 10, LYST, SLC38A8)*,
- hemizygous pathogenic or likely pathogenic HGMD-listed variants in *GPR143* or *FRMD7*,
- the *TYR*:c.[− 301C;575 C > A;1205 G > A] or *TYR*:c.[− 301C;575 C;-1205 G > A] albinism-related haplotype[19] in homozygous state
- the *TYR*:c.[− 301C;575 C > A;1205 G > A] or *TYR*:c.[− 301C;575 C;-1205 G > A] albinism-related haplotype in heterozygous state plus a heterozygous pathogenic or likely pathogenic variant in *TYR*.

We also excluded from our analysis individuals who were found to carry pathogenic or likely pathogenic variants in the *TYR* or *OCA2* gene (in a heterozygous state). Small-scale genetic changes (with the exception of *TYR*:c.1205 G > A and *OCA2*:c.1327 G > A), copy number and structural variants were factored in.

*Common albinism-associated alleles:* The effect of one of the studied changes (*TYR* c.1205 G > A) has been previously shown to be modified by the following two variants[19–21,25,34–36] which have been added as a covariate to the regression model:

- *TYR* c.− 301C > T [rs4547091], located in the *TYR* promoter; the derived, non-ancestral allele c.− 301C (or c.− 301=) has a MAF of ~ 60% in non-Finnish European (NFE) populations in the Genome Aggregation Database (gnomAD v2.1.1)[33];
- *TYR* c.575 C > A (p.Ser192Tyr) [rs1042602], which has an MAF of ~ 36% in gnomAD NFE populations.

It is highlighted that the *TYR* c.− 301= and the *TYR* c.575 C > A alleles are frequently found in linkage disequilibrium, so the presence of the latter would, in most cases, suggest the presence of the former on the same haplotype.

**Case-control comparisons between *TYR/OCA2* genotype groups.** Our primary analysis focused on performing case-control comparisons between groups of genotypes that include different alleles at positions *TYR*:c.1205 and *OCA2*:c.1327. These groups were:

- group A (reference group): homozygous for *TYR*:c.1205 = and *OCA2*:c.1327 =
- group B (single TYR heterozygote group): heterozygous for *TYR*:c.1205 G > A and homozygous for *OCA2*:c.1327 =.
- group C (single OCA2 heterozygote group): homozygous for *TYR*:c.1205 G = and heterozygous for *OCA2*:c.1327 G > A.
- group D (dual heterozygote group): heterozygous for the *TYR*:c.1205 G > A and *OCA2*:c.1327 G > A variant combination.

Analyses were also conducted in individuals who were found to be homozygous for *TYR*:c.1205 G > A (but are considered unsolved, i.e.

they do not also carry the *TYR* c.575 C > A or the *TYR* c.− 301 = variant in the homozygous state); these were split into two groups:

- *group E:* homozygous for *TYR*:c.1205 G > A and *OCA2*:c.1327 =;
- *group F:* homozygous for *TYR*:c.1205 G > A and heterozygous for *OCA2*:c.1327 G > A.

Group A was used as the reference group to which all other groups were compared using Firth regression analysis (*i.e.* the odds ratio for this group was fixed at 1).

The primary analysis focused on a mixed case cohort (including 173 probands from the University Hospital of Bordeaux cohort and 31 cases from the Genomics England 100,000 Genomes Project dataset) and a control cohort from the Genomics England 100,000 Genomes Project dataset (20,350 unrelated individuals) (Fig.2, Supplementary Fig. 1a and Supplementary Table 2). A secondary analysis focusing only on individuals from the Genomics England 100,000 Genomes Project dataset (31 cases, 20,350 controls) was also undertaken (Supplementary Fig. 1b and Supplementary Table 8).

### Replication of findings in the UK Biobank cohort

The effect of dual heterozygosity for *TYR*:c.1205 G > A and *OCA2*:c.1327 G > A was studied in the UK Biobank. UK Biobank is a biomedical resource containing in-depth genetic and health information from > 500,000 individuals from across the UK[40]. A subset of UK Biobank volunteers underwent enhanced phenotyping including visual acuity testing (131,985 individuals) and OCT imaging of the central retina (84,748 individuals)[61]. All UK Biobank volunteers whose data were analysed as part of this study were imaged using the 3D OCT-1000 Mark II device (Topcon, Japan); the relevant methodology has been previously described[61]. Notably, the diagnosis of albinism has been assigned to 39 UK Biobank participants (ICD-10 term "Albinism" [E70.3] in data field 41270 or resource 591) (Supplementary Table 10).

Aiming to assess the robustness of the results in the combined University Hospital of Bordeaux and Genomics England 100,000 Genomes Project dataset, we performed case-control analyses in UK Biobank volunteers. We focused on individuals who were estimated by principal component analysis to have European-like ancestries (data field 22006). First, genotyping array data were used to obtain genotypes for the *TYR*:c.1205 G > A variant and *OCA2*:c.1327 G > A variant (data field 22418 including information from the Applied Biosystems UK Biobank Axiom Array containing 825,927 markers). Only individuals reliably genotyped at both sites were retained. Aiming to reduce the likelihood of obtaining spurious signals due to the presence of albinism-related variants other than the two studied changes (*TYR*:c.1205 G > A and *OCA2*:c.1327 G > A), we excluded UK Biobank participants whose exome sequencing data (data field 23157) suggested that they carried at least one HGMD-listed variant in an albinism-related gene.

We used Firth logistic regression to perform case-control comparisons between genotype groups A–D. Sex and the number of common albinism-associated alleles were used as covariates (Fig.3a, Supplementary Fig. 2 and Supplementary Table 3).

Given that reduced visual acuity and increased central retinal thickness (due to underdevelopment of the fovea) are two hallmark features of albinism, we investigated the impact of selected *TYR/OCA2* variant combinations on these two quantitative traits. Aiming to reduce the likelihood of obtaining spurious signals due to the presence/impact of ophthalmic conditions/features not related to albinism, we excluded all UK Biobank volunteers that did not have the ICD-10 term "Albinism" [E70.3] and were assigned an ophthalmology-related ICD-10 code ("Chapter VII Diseases of the eye and adnexa").

We first obtained data on the left LogMAR visual acuity for UK Biobank volunteers (data field 5201, "instance 0" datasets). These visual acuity measurements were then used to compare visual performance between groups of volunteers with selected *TYR/OCA2*

genotype combinations (genotype groups A–D) (Supplementary Fig. 3). As the obtained distributions deviated from normality, the Kruskal-Wallis test was used. Pair-wise comparisons were performed, and the *p*-values were adjusted using the Benjamini-Hochberg method (Fig.3b and Supplementary Table 4). Additional analyses using a linear regression approach were also conducted; sex and the number of common albinism-associated alleles were included as covariates (Supplementary Table 5).

We then obtained central retinal thickness measurements from UK Biobank OCT images using the "macular thickness at the central subfield" data field (27802; defined as the average distance between the hyperreflective bands corresponding to the RPE and the internal limiting membrane (ILM), across the central 1 mm diameter circle of the ETDRS grid) for UK Biobank volunteers[62]. The obtained measurements were subsequently used to compare central retinal thickness between groups of UK Biobank volunteers with selected *TYR/OCA2* genotype combinations (groups A–D). As some of the obtained distributions deviated from normality, the Kruskal-Wallis test was used. Pair-wise comparisons were performed, and the *p*-values were adjusted using the Benjamini-Hochberg method (Fig.3c and Supplementary Table 4). Additional analyses using a linear regression approach were also conducted; sex and the number of common albinism-associated alleles were included as covariates (Supplementary Table 6).

It is noted that we opted to perform all UK Biobank analyses on left eye data. This is because, as per study protocol[61], the left eye was tested after the right eye, potentially making the measurements from the left side less prone to artefacts (as the participants were more familiar with the test).

### Additional analysis in the University Hospital of Bordeaux albinism cohort

Further analyses were undertaken using only data from individuals in the University Hospital of Bordeaux albinism cohort. After excluding study subjects who did not report having European-like ancestries, the cohort was split into cases with and cases without a molecular diagnosis of albinism (*i.e.* "solved" and "unsolved"). Individuals without a molecular diagnosis who carried heterozygous pathogenic or likely pathogenic variants in the *TYR* or *OCA2* gene (with the exception of the two studied changes *TYR*:c.1205 G > A and *OCA2*:c.1327 G > A) were then removed from the unsolved group. The patterns observed in genotype groups A–D were then assessed using both contingency table methods and Firth regression analyses (Supplementary Table 9).

To reduce the likelihood that the phenotypic findings in the *TYR*:c.1205 G > A and *OCA2*:c.1327 G > A dual heterozygotes are not linked to albinism, additional genetic testing was initiated in the relevant 9 cases from the University Hospital of Bordeaux albinism cohort (Supplementary Data 3). This included analysis of the following key genes: *PAX6, NYX, CACNA1F, GRM6, GNAT1, RHO, PDE6B, TRPM1, GPR179, SLC24A1, LRIT3, GNB3, GUCY2D, CNGB3, CNGA3, PDE6C, GNAT2, PDE6H, ATF6*.

### Reporting summary

Further information on research design is available in the Nature Portfolio Reporting Summary linked to this article.

## Data availability

To avoid compromising the privacy of research participants, key raw data used in this study are only available via controlled access. The Genomics England 100,000 Genomes Project dataset is available under restricted access through a procedure described at https://www.genomicsengland.co.uk/about-gecip/for-gecip-members/data-and-data-access. The UK Biobank dataset is available under restricted access through a procedure described at http://www.ukbiobank.ac.uk/using-the-resource/. Data from the University Hospital of Bordeaux albinism cohort have been partly reported in previous publications

from our group[19,22]. Additional individual-level phenotypic information and raw genotypic data from this study cannot be deposited in publicly available resources because the contributing institutions do not have consent for this. Access to these data (e.g., for the purpose of verifying the research in this article) is contingent on appropriate ethics approval and data-sharing agreements. Beginning 3 months and ending 24 months after article publication, all relevant data access requests should be made to the corresponding author. Responses to valid requests will be reasonably attempted and initiated within a month. All other data supporting the findings of this study are available within the article (including its Supplementary Information files).

## Code availability

The scripts used to analyse the datasets included in this study are available at https://github.com/davidjohngreen/tyr-oca2.

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

## Acknowledgements

We acknowledge the following sources of funding: the Wellcome Trust (224643/Z/21/Z, Clinical Research Career Development Fellowship to P.I.S.; 200990/Z/16/Z, Transforming Genetic Medicine Initiative to G.C.B.); the UK National Institute for Health Research (NIHR) Clinical Lecturer Programme (CL-2017-06-001 to P.I.S.); the NIHR Manchester Biomedical Research Centre (NIHR 203308 to P.I.S. and G.C.B.); Retina UK and Fight for Sight (GR586, RP Genome Project - UK Inherited Retinal Disease Consortium to G.C.B.); Christopher Green (D.J.G.); the French Albinism Association (Genespoir) and the French National Research Agency (Agence Nationale de la Recherche; ANR-21-CE17-0041-01 to B.A.). The UK Biobank Eye and Vision Consortium is supported by funding from the NIHR Biomedical Research Centre at Moorfields Eye Hospital and UCL Institute of Ophthalmology, the Alcon Foundation and the Desmond Foundation. The complete list of members of this Consortium can be found in the Supplementary Information. This research was made possible through access to the data and findings generated by the 100,000 Genomes Project. The 100,000 Genomes Project is managed by Genomics England Limited (a wholly owned company of the Department of Health and Social Care). The 100,000 Genomes Project is funded by the NIHR and NHS England. The Wellcome Trust, Cancer Research UK, and the Medical Research Council have also funded research infrastructure. The 100,000 Genomes Project uses data provided by patients and collected by the National Health Service (NHS) as part of their care and support. We acknowledge the contribution of the Genomics England Research Consortium to the 100,000 Genomes Project. The complete list of members of this Consortium can be found in the Supplementary Information. We thank Dominique Bonneau, Dominique Bremond-Gignac, Benjamin Dauriat, Sabine Defoort, Hélène Dollfus, Isabelle Drumare, Mélanie Fradin, Jamal Ghoumid, Anne-Marie Guerrot, Smail Hadj Rabia, Elsa Laumonier Demory, Guylène Le Meur, Sylvie Manouvrier, Isabelle Meunier, Fanny Morice-Picard, Valérie Pelletier, Audrey Putoux, Vasily Smirnov, Catherine Vincent-Delorme, Xavier Zanlonghi, Catherine Yardin for their valuable input and assistance with phenotypic data collection. Lastly, we acknowledge the help of Cécile Courdier at the University Hospital of Bordeaux Molecular Genetics Laboratory, Sophie Javerzat at the University of Bordeaux, Dave Gerrard at the University of Manchester, and Claire Hardcastle, Chris Campbell, Steph Barton and Simon Ramsden at the North West of England Genomic Laboratory Hub.

## Author contributions

P.I.S. conceived and designed the experiments. The UK Biobank Eye and Vision Consortium, T.F., E.B., B.A., G.C.B. and P.I.S. provided datasets and analytical tools. D.J.G., V.M., E.L., C.P., T.F., B.A. and P.I.S. analysed the data. P.I.S. wrote the manuscript with support from D.J.G. All authors critically revised and approved the manuscript. G.C.B. and B.A. contributed equally to this work.

## Competing interests

E.B. is a paid consultant and equity holder of Oxford Nanopore, a paid consultant to Dovetail, and a non-executive director of Genomics England, a limited company wholly owned by the UK Department of Health and Social Care. All other authors declare no competing interests.

## Additional information

---

## UK Biobank Eye and Vision Consortium

**Graeme C. Black**[1,5,7] **& Panagiotis I. Sergouniotis** [1,4,5,6] ✉

