## [Peer Review File · Nature Communications]

The co-occurrence of genetic variants in the TYR and OCA2 genes confers susceptibility to albinismREVIEWER COMMENTS

Reviewer #1 (Remarks to the Author):

This manuscript by Green at all utilizes large population datasets from individuals with albinism and the 100k_GP cohort. They find a risk for dual heterozygosity for two fairly common variants in the TYR and OCA2 genes. While this is intriguing, certain features need to be clarified to demonstrate this phenomenon.

1. The results and Figure 1 are a bit confusing - 1121 patients with albinism are recruited, and the figure indicates inclusion is based on the absence of rare, albinism-implicated variants. This should be clarified on several levels in the figure and text. Firstly, does this alter the number of individuals in the case-control analysis? This number seems quite large already for a rare disease. Secondly, surely there are carriers of pathogenic variants among the cases and controls, which implies they should be removed by the criterion as stated? Thirdly, the OCA2 c.1327G>A could be considered rare for recessive disease, and TYR c.1205G>A is implicated in albinism in cis with p.Ser192Tyr. Perhaps this should also be corrected to represent "no albinism-implicated biallelic genotypes."
2. Molecular diagnostic testing of less than or equal to 19 genes may miss a small percentage of molecular diagnoses. What is the diagnostic yield of this test, and how does this compare to published studies including exome and genome sequencing? Did all individuals have CNV analysis performed?
3. If those in the albinism group with clear monogenic diagnosis are removed, then this is also a good control group for an additional analysis. If dual heterozygosity is a causal genotype for albinism, one would predict that individuals with albinism without pathogenic genotypes would be enriched for this dual heterozygosity mechanism compared to those with albinism due to other genotypes.
4. Dual heterozygosity conferring risk for albinism delineates this from the concept of digenic causal genotypes as have been proposed, or perhaps explains them better. However, it is perhaps improper to draw comparison to risk of breast cancer and other adult onset diagnoses caused by defined risk alleles. Breast cancer is a clear diagnosis that can be ascertained at different ages (assuming screening). Albinism is a congenital disorder that has very little phenotypic modification past childhood. And it is possible that those in the control cohort have mild phenotypes of albinism and are simply not ascertained, as those with unreported or undiagnosed conditions exist in large population databases. This should be listed as a caveat.
5. Figure 2 - the "n" for each group should be provided. Within this comparison, the group of dual homozygotes should have the highest risk for albinism, if dual heterozygosity confers risk for this phenotype, and therefore be present at very low levels in the control population. It would be best to assess this group directly instead of grouping them with the homozygous;heterozygous genotypes. If a small number of individuals, the clinical details should be provided. It would be interesting to know more about the phenotypes, including iris transillumination and grade of foveal hypoplasia.
6. Endophenotypes of albinism and risk of albinism seem at odds. Is the argument that they are on the mild end of the spectrum and this is extreme end of variable expressivity, or that there is incomplete penetrance. Given there is clinical data available on the albinism group, this should be evaluated more deeply. In particular, the range of phenotypes for the dual heterozygotes should be clustered towards the milder end of the spectrum of endophenotypes.
7. If dual heterozygosity is true, then autosomal dominantly inherited albinism would be more commonly reported given the vast literature. In most cases of apparent dominant transmission, this is due to triallelic inheritance where one parent is affected (biallelic variants in one gene) and the other is a carrier in the same gene. It's also likely that endophenotypes would be detected more frequently in the albinism group. Analyzing the the clinical data available in the albinism group of dual heterozygotes would be very valuable, including pedigree analysis and segregation of variants when possible. Pedigree assessment coupled with deep phenotyping would be the proper way to define risk if sufficient families are available.
8. Other considerations should be given for what this risk means and alternative explanations. The detected risk of dual heterozygotes could be of a cryptic pathogenic TYR or OCA alleles in trans (or in cis, aside from the 2 covariate variants for p.Arg402Gln). However, the molecular data in the case cohort is quite limited. Or there could be fundamental population differences in the albinism group and

the control group not detected by the 35 nucleotides assessed, given that admixture may confound this analysis, and since both variants exhibit broad differences in subpopulation frequency. These issues need to be addressed more broadly, along with proposed future directions.

Reviewer #2 (Remarks to the Author):

Green et al. propose that a frequent (minor allele frequency in population is 0.27) heterozygous TYR hypomorphic variant/SNP (c.1205G>A) shows a synergistic effect when together with a heterozygous known causal and moderately frequent (minor allele frequency 0.003) OCA2 variant (c.1327G>A), to increase the risk of ocular albinism and the underlying endophenotypes.

This paper addresses an important and understudied area, i.e. to unravel how frequent and rare variants in different genes may underlie complex inheritance. Ocular albinism, because of the involvement of many different genes carrying rare, moderately rare and frequent variants, could represent a very nice model to improve our understanding of complex inheritance. In this case the interplay between variants in two genes was studied. The paper is well written, and I appreciate the comments in the discussion in which they are careful to consider what the identified odds ratios mean, i.e. an increased risk to develop ocular albinism, and that other unidentified factors also play a role.

I do have concerns on the study set-up that could have a big impact on the conclusions.

Major concerns:

Two alleles in one locus and two alleles in another locus result in a maximum of 9 different allele combinations. The authors arranged these combinations in 6 groups in their statistical analysis, but it is not clear why this was done. From a statistical point of view, a completely unbiased approach would be to look at all combinations separately. For example, in Table S2, the ratio for the combination AG (TYR)/AG (OCA2) in group C (15 cases/129 controls) is very similar to the ratio for GG/AG in group B (13 cases/150 controls), but this is masked by the fact that group B also contains the combination AG/GG (105 cases/8250 controls). The same is true for other comparisons.

If one sums up the cases and controls for the three TYR combinations, the cases/controls ratios are similar (GG, 0.019; AG, 0.015, AA, 0.030), whereas the OCA2 groups are, not unexpected based on the highly penetrant c.1327G>A variant, very different (GG, 0.016; AG, 0.096; AA, 3.0). It seems as if the presumed TYR/OCA2 synergistic effect for the combination AG/AG can be fully explained by the presence of the A in the AG of OCA2.

An important point to consider is that the OCA2 c.1327G>A variant, when present in a homozygous manner, acts as a fully penetrant causal allele, whereas the TYR c.1205G>A variant clearly is a hypomorphic allele. Is it thus logical to test an hypothesis that they have 'equal weight' effects, when both present in a heterozygous manner? Can the authors elaborate on this?

A synergistic effect would mean that small, subclinical effects of the alleles on their own, when present together, give rise to an effect that is bigger than the effects of each allele separately. Based on previous studies, there does not seem to be a direct interaction between the TYR and OCA2 proteins. Irrespective of an additive or synergistic mechanism of disease, would it not make sense that albinism cases carrying TYR c.1205G>A and another causal TYR variant in trans, would have a more severe phenotype (i.e. severe endophenotypes, such as retinal thickness and central visual acuity) if the OCA2 c.1327G>A is present? Is there evidence for this? As the latter is quite rare, this may be difficult to study. But the converse also could be true. If an albinism case carries the OCA2 c.1327G>A variant with another OCA2 variant in trans, the presence of the TYR c.1205G>A could also give rise to a more severe phenotype. This kind of data could be presented in Supplementary Table 4, but unfortunately,

the data for groups B and D were lumped together. The underlying endophenotype data for the subgroups should be provided.

Minor points:

In literature there is some controversy over the causal nature of the TYR c.1205G>A variant, but it very likely acts as a hypomorphic allele when together with a severe TYR allele. The high allele frequency of TYR c.1205G>A in European populations is striking – is reduced penetrance observed in individuals with TYR c.1205G>A and a second allele? The authors could provide further discussion that the TYR c.1205G>A variant may be of low penetrance and the high allele frequency observed could have a cumulative effect on the overall risk of albinism.

The introduction could benefit from some additional information into the previous reported variants implicated in albinism (from line 89) to introduce the frequent genes and variants that are reported in the methods section. E.g. line 237: c.-301 TYR variant is mentioned, but it is not apparent until line 316 the relevance of this. Likewise, c.575C>A/p.Ser192Tyr was found more often in cis with c.1205G>A in compound heterozygous cases, than seen in the general population. Although there does not seem to be an added effect of c.575C>A on the phenotype in its presence, the authors may mention the existence of this frequent variant.

The authors may also mention the Burkard et al 2018 CNGA3/CNGB3 digenic triallelic inheritance.

Supplementary data 1: Nomenclature of deletions is not according to HGVS

Why are synonymous variant such as HPS5:c.219G>A (p.Arg73=) and SLC45A2:c.1518C>T (p.Val506=) provided? What is the evidence that they are causal?

Please check all tables and figures to capitalize the headings for consistency (e.g. in Supplementary Data 1: 'rare single nucleotide...' should be 'Rare single nucleotide...', and 'male'/'female' in column B should be capitalized, etc.

Italicize gene names in Supplementary Data 1 and 2 tables

Line 130: the minor allele frequencies for both variants are reported. This sentence could be better linked to the previous sentence.

Reviewer #3 (Remarks to the Author):

This paper by Green and al. investigates the epistatic effect of two mutations already shown to have a main effect on albinism. These mutations are in two genes which were possibly interacting, and thus there is a strong prior for possible interaction.

Case control design is comparing albinism cases mainly of French origin and controls from Britain. While they are both from European descent, we could expect differences that could lead to false positives – maybe even more for rare variants. The authors try to account for this design. Interestingly, this study finds a very strong association of double heterozygotes with albinism condition in this heterogeneous case controls design and some association with endophenotypes within the homogenous British population, which is a bit reassuring. One of the main questions I have is how much of this association is due to main effects rather than to pure epistasis.

1/ I am not sure what the difference is between qualitative and quantitative approach. Testing interaction is estimated quantitatively (interactions test, likelihood models ...). While not crucial, I do

not see why the approach used by the authors is declared quantitative as opposed to qualitative ones.

2/ I am not sure what is the difference, in essence, between the two analyses. The first one is a logistic regression while the other one is comparing various groups of (di-genotypes). Analysis 2 can fit into logistic regression so this is just an unbalanced way to represent the analyses. Maybe better say that analysis one is « linear » in the sense that they are testing association of the number of rare alleles and their product (if I understand well whatr they call logistic regression).

3/ Line 212 : I find the approach description lacks a bit of clarity. What are targeted analyses ? It is clarified later on, but explaining in the main text how this could help correcting for possible stratification.

4/ Why are you only using 35 « supposedly neutral » to estimate the inflation factor. This inflation factor 1.04 seems a bit low for a study comparing cases and controls from two countries (high prior of false positive due to stratification) : this is a strong assumption, which may be true but really needs strong support, with as much SNPs as possible.

5/ I am not convinced that the genetic stratification is corrected for and propose to check the allele frequency of both the genetic variants and combination of genotypes in a French reference population.

6/ It is not clear how the interaction model is tested. The authors claim that there is interaction but it tests alleles which are already found to increase the risk themselves. Again, please clarify the exact model which is being tested - for instance tease apart the beta and pvalue for main and interaction effects. I well noticed that the increase in OR on Figure 2 seems non linear, but we'd need a more formal testing as the sample size is small.

Panos Sergouniotis FRCOphth, PhD
Manchester Centre for Genomic Medicine
Manchester University NHS Foundation Trust,
Oxford Road, Manchester M13 9WL, UK

email: panagiotis.sergouniotis@manchester.ac.uk

July 14th, 2023

To whom it may concern

Re: Revision of NCOMMS-22-51271 (“The co-occurrence of genetic variants in the TYR and OCA2 genes confers susceptibility to albinism”)

Dear Sir/Madan

Please find below a point-by-point response to the reviewers’ comments relating to our manuscript entitled: “*The co-occurrence of genetic variants in the TYR and OCA2 genes confers susceptibility to albinism*”. We are grateful to the 3 reviewers for the helpful feedback. We have now performed extensive re-analyses and included additional data in line with their recommendations. These have improved the manuscript and strengthened the conclusions (for example the odds ratio for the studied dual heterozygous genotype increased from >4.4 to >6.8).

REVIEWER 1 COMMENTS

This manuscript by Green at all utilizes large population datasets from individuals with albinism and the 100k_GP cohort. They find a risk for dual heterozygosity for two fairly common variants in the TYR and OCA2 genes. While this is intriguing, certain features need to be clarified to demonstrate this phenomenon.

Comment #1.1: *1. The results and Figure 1 are a bit confusing - 1121 patients with albinism are recruited, and the figure indicates inclusion is based on the absence of rare, albinism-implicated variants. This should be clarified on several levels in the figure and test. Firstly, does this alter the number of individuals in the case-control analysis? This number seems quite large already for a rare disease. Secondly, surely there are carriers of pathogenic variants among the cases and controls, which implies they should be removed by the criterion as stated? Thirdly, the OCA2 c.1327G>A could be considered rare for recessive disease, and TYR c.1205G>A is implicated in albinism in cis*

with *p.Ser192Tyr*. Perhaps this should also be corrected to represent "no albinism-implicated biallelic genotypes."

Response #1.1: In response to comments from all three reviewers (including comments #1.2, #2.1, #2.4, #3.1, #3.2 and #3.5), we have now amended and clarified the inclusion criteria and the analytical approach. Figure 1 has been redrawn and the updated version can be found below:

Comment #1.2: 2. Molecular diagnostic testing of less than or equal to 19 genes may miss a small percentage of molecular diagnoses. What is the diagnostic yield of this test, and how does this compare to published studies including exome and genome sequencing? Did all individuals have CNV analysis performed?

Response #1.2: The diagnostic yield of the clinical genetic test used in the Bordeaux cohort was 75%. This was higher than the diagnostic rate described in other studies of unselected albinism cohorts including:

- Chan *et al.*, Genes 2023 [DOI: 10.3390/genes14010135]: 70% (US; panel-based testing)

- Jackson *et al. Am J Med Genet C Semin Med Genet* 2020 [DOI: 10.1002/ajmg.c.31837]: 56% (UK; whole genome sequencing)
- Chan *et al., Genes* 2021 [DOI: 10.3390/genes12040508]: 42% (UK; whole genome sequencing or panel-based testing)
- Mauri *et al. J Hum Genet* 2017 [DOI: 10.1038/jhg.2016.123]: 74% (Italy; panel-based testing)

Following the reviewer's comment, we performed extensive copy number variant (CNV) and structural variant analyses in all study subjects. Incorporating these classes of variation has strengthened the detected signals.

Comment #1.3: *3. If those in the albinism group with clear monogenic diagnosis are removed, then this is also a good control group for an additional analysis. If dual heterozygosity is a causal genotype for albinism, one would predict that individuals with albinism without pathogenic genotypes would be enriched for this dual heterozygosity mechanism compared to those with albinism due to other genotypes.*

Response #1.3: We thank the reviewer for this recommendation. We have now performed additional analyses comparing 'solved' with 'unsolved' cases in the Bordeaux albinism cohort. As predicted, individuals with albinism and no pathogenic genotypes are enriched for dual heterozygosity involving *TYR* c.1205G>A and *OCA2* c.1327G>A (p-value=0.0078). The results of this analysis are presented in a new Table (Supplementary Table 7).

Comment #1.4: *4. Dual heterozygosity conferring risk for albinism delineates this from the concept of digenic causal genotypes as have been proposed, or perhaps explains them better. However, it is perhaps improper to draw comparison to risk of breast cancer and other adult onset diagnoses caused by defined risk alleles. Breast cancer is a clear diagnosis that can be ascertained at different ages (assuming screening). Albinism is a congenital disorder that has very little phenotypic modification past childhood. And it is possible that those in the control cohort have mild phenotypes of albinism and are simply not ascertained, as those with unreported or undiagnosed conditions exist in large population databases. This should be listed as a caveat.*

Response #1.4: We have now removed the sentence where we highlight the similarity of the detected odds ratio to that found for disease-related variants associated with breast cancer and glaucoma (P9).

Comment #1.5: *5. Figure 2 - the "n" for each group should be provided. Within this comparison, the group of dual homozygotes should have the highest risk for albinism, if dual heterozygosity*

confers risk for this phenotype, and therefore be present at very low levels in the control population. It would be best to assess this group directly instead of grouping them with the homozygous;heterozygous genotypes. If a small number of individuals, the clinical details should be provided. It would be interesting to know more about the phenotypes, including iris transillumination and grade of foveal hypoplasia.

Response #1.5: All numerical data (including for individual groups) are provided in the Supplementary Tables. In response to this comment and recommendations from the other two reviewers (e.g. in comments #2.1 and #3.2) we have now altered the analytical approach and no longer lump different genotypes into mixed groups.

With regards to ‘dual homozygosity’: no individuals with this genotype were detected in the control population and only two individuals with this were found in the case cohort. Clinical details in these two cases can now be found in Supplementary Data 3.

Comment #1.6: *6. Endophenotypes of albinism and risk of albinism seem at odds. Is the argument that they are on the mild end of the spectrum and this is extreme end of variable expressivity, or that there is incomplete penetrance. Given there is clinical data available on the albinism group, this should be evaluated more deeply. In particular, the range of phenotypes for the dual heterozygotes should be clustered towards the milder end of the spectrum of endophenotypes*

Response #1.6: Incomplete penetrance and variable expressivity can be viewed as closely linked biological phenomena that lead to non-linearity between genotype and phenotype. When there is an informative, well-defined biomarker associated with a phenotype of interest (e.g. HbA1c in the case of MODY) the distinction between incomplete penetrance and variable expressivity can appear artificial. In the context of albinism, foveal hypoplasia can be viewed as a key quantitative biomarker. Notably, many genetically confirmed cases of albinism that are at the milder end of the clinical spectrum present only with low-grade foveal hypoplasia and may experience considerable delay in receiving a diagnosis. In these cases, non-genetic parameters, for example the probability of the affected individual interacting with a suitably qualified ophthalmic practitioner (e.g. optician or ophthalmologist) can be strongly influential.

Overall, we view albinism more in the context of variable expressivity than of incomplete penetrance. The term “risk of albinism” was mainly used as shorthand for “the probability of receiving a clinical diagnosis of albinism” and we have pointed to this in the Abstract (P3, L49), the legend of Figures 2 and 3 (P10; P12), the Main Text (P8, L140), the Methods (P17, L307) and the Supplementary Information (P2,P4,P5,P9).

With regards to the inclusion of phenotypic information, we have now added a new Table with clinical data on dual heterozygotes (n=10) and dual homozygotes (n=2) (Supplementary Data 3). We note however that a detailed characterization/comparison of the clinical presentations in the different genotype groups that we have studied is outside the scope of this study.

Comment #1.7: *7. If dual heterozygosity is true, then autosomal dominantly inherited albinism would be more commonly reported given the vast literature. In most cases of apparent dominant transmission, this is due to triallelic inheritance where one parent is affected (biallelic variants in one gene) and the other is a carrier in the same gene. It's also likely that endophenotypes would be detected more frequently in the albinism group. Analyzing the clinical data available in the albinism group of dual heterozygotes would be very valuable, including pedigree analysis and segregation of variants when possible. Pedigree assessment coupled with deep phenotyping would be the proper way to define risk if sufficient families are available.*

Response #1.7: As the reviewer highlights:

- (i) it is not uncommon for the parents of people with albinism to have features of the condition. A recent study for example found that 1 in 3 parents of children with albinism have a degree of foveal hypoplasia (Lejoyeux *et al.*, *Ophthalmic Genet.* 2022 [DOI: 10.1080/13816810.2022.2121841]).
- (ii) the clinical features that we describe here as albinism endophenotypes (foveal hypoplasia and reduced visual acuity) are very frequently detected in people with albinism. A study of 522 individuals with albinism for example found that 99.3% of cases had foveal hypoplasia (Kruijt *et al.* *Ophthalmology.* 2018 [DOI: 10.1016/j.opthta.2018.08.003]).

We agree with the reviewer that clinical data analysis and pedigree assessment studies would be of interest and, as mentioned in response #1.6, relevant additional data have now been included in Supplementary Data 3. However, an in-depth discussion of clinical information in the various genotype groups is outside the scope of this study. Notably, we have decided to conduct large-scale analyses that focused on populations rather than specific families in order to increase the robustness and generalisability of our findings.

Comment #1.8: *8. Other considerations should be given for what this risk means and alternative explanations. The detected risk of dual heterozygotes could be of a cryptic pathogenic TYR or OCA alleles in trans (or in cis, aside from the 2 covariate variants for p.Arg402Gln). However, the molecular data in the case cohort is quite limited. Or there could be fundamental population differences in the albinism group and the control group not detected by the 35 nucleotides assessed, given that admixture may confound this analysis, and since both variants exhibit broad*

differences in subpopulation frequency. These issues need to be addressed more broadly, along with proposed future directions.

Response #1.8: We have now expanded the discussion of the caveats in the penultimate paragraph of the manuscript (P13-14, L197-217):

“One limitation of this study is our inability to match stringently the albinism cases with the unaffected controls, especially in terms of genetic background.⁴³ Although recent ancestry was considered, our analysis was imperfect as it was not possible to reliably assign genetic ancestry to most albinism cases. We used a combination of orthogonal approaches to evaluate the robustness and generalisability of our findings. First, we used 35 presumed neutral single-nucleotide variants to calculate the genomic inflation factor lambda (λ_{GC})^{44,45}; the λ_{median} was found to be 1.04, in keeping with limited confounding by ancestry (Supplementary Table 5). Subsequently, we performed a targeted secondary analysis using only Genomics England 100,000 Genomes Project data which allowed us to study case and control groups of high genetic similarity. The results supported our key findings and increased confidence in the validity of the detected associations (Supplementary Fig.1b and Supplementary Table 6). Another potential caveat is that the detected signal in dual heterozygotes for TYR:c.1205G>A and OCA2:c.1327G>A could, to a degree, be driven by undetected pathogenic variation in either TYR or OCA2. This cannot be excluded despite the fact that medical-grade genetic analyses were undertaken. It can be argued however that significant confounding is unlikely especially given that the effect of dual heterozygosity for TYR:c.1205G>A and OCA2:c.1327G>A appears to be greater than that of genotypes involving heterozygosity for either TYR:c.1205G>A or OCA2:c.1327G>A alone (Supplementary Table 2, 3, 6 and 7).”

We would like to mention that we respectfully disagree with the reviewer’s comment that ‘the molecular data in the case cohort is quite limited’. In addition to the information provided in response to comment #1.2, we would like to highlight that the genetic analysis in the Bordeaux case cohort was comprehensive and was undertaken in a national reference laboratory that has over 20 years of experience with genetic testing for albinism (P15, L233-235).

With regards to potential issues with population differences between the case and control groups. We agree this could be a potential confounder and acknowledge this as a caveat. It is noted however that the main findings were replicated in secondary analyses conducted in targeted subgroups in which there was significant genetic similarity between cases and controls such as:

- (i) *Genomics England 100,000 Genomes project*: the detected odds ratio for dual heterozygosity was 14.8 (compared to 14.0 in the primary analysis) (Supplementary Fig.2b; Supplementary Table 6);
- (ii) *UK Biobank*: the detected odds ratio for dual heterozygosity was 16.3 (Fig.3a; Supplementary Table 3);
- (iii) *Bordeaux albinism cohort* (as per comment #1.3) (Supplementary Table 7).

REVIEWER 2 COMMENTS

Green et al. propose that a frequent (minor allele frequency in population is 0.27) heterozygous TYR hypomorphic variant/SNP (c.1205G>A) shows a synergistic effect when together with a heterozygous known causal and moderately frequent (minor allele frequency 0.003) OCA2 variant (c.1327G>A), to increase the risk of ocular albinism and the underlying endophenotypes.

This paper addresses an important and understudied area, i.e. to unravel how frequent and rare variants in different genes may underlie complex inheritance. Ocular albinism, because of the involvement of many different genes carrying rare, moderately rare and frequent variants, could represent a very nice model to improve our understanding of complex inheritance. In this case the interplay between variants in two genes was studied. The paper is well written, and I appreciate the comments in the discussion in which they are careful to consider what the identified odds ratio's mean, i.e. an increased risk to develop ocular albinism, and that other unidentified factors also play a role. I do have concerns on the study set-up that could have a big impact on the conclusions.

Major concerns:

Comment #2.1: *Two alleles in one locus and two alleles in another locus result in a maximum of 9 different allele combinations. The authors arranged these combinations in 6 groups in their statistical analysis, but it is not clear why this was done. From a statistical point of view, a completely unbiased approach would be to look at all combinations separately. For example, in Table S2, the ratio for the combination AG (TYR)/AG (OCA2) in group C (15 cases/129 controls) is very similar to the ratio for GG/AG in group B (13 cases/150 controls), but this is masked by the fact that group B also contains the combination AG/GG (105 cases/8250 controls). The same is true for other comparisons.*

Response #2.1: We thank the reviewer for this comment. Following the above recommendation and aiming for an less biased approach, we have now re-analysed our data looking at each genotype separately. The findings strengthen our results and are presented in the updated Fig.1, Fig.2, Supplementary Fig.1 and Supplementary Tables 2, 3 and 6.

With regards to the similarity between the combination AG (TYR)/AG(OCA2) (dual heterozygotes) and GG(TYR)/AG(OCA2) (OCA2 heterozygotes): the incorporation of CNV analysis to our pipeline (following a suggestion by reviewer #1) has provided some clarity. A detailed discussion on this can be found in our response to the related comment #2.2 below.

Comment #2.2: *If one sums up the cases and controls for the three TYR combinations, the cases/controls ratios are similar (GG, 0.019; AG, 0.015, AA, 0.030), whereas the OCA2 groups are, not unexpected based on the highly penetrant c.1327G>A variant, very different (GG, 0.016; AG,*

0.096; AA, 3.0). It seems as if the presumed TYR/OCA2 synergistic effect for the combination AG/AG can be fully explained by the presence of the A in the AG of OCA2.

Response #2.2: We are grateful to the reviewer for highlighting this key issue.

In response to comment #1.2 by reviewer #1, we have now incorporated CNV and structural variant analysis to our pipeline. As a result, many cases assigned to the GG(TYR)/AG(OCA2) (OCA2 heterozygotes) group are now considered solved. This has significantly altered the case/control ratios. In the Table below we have summarised a set of observations that do not support the statement that ‘the presumed TYR/OCA2 synergistic effect for the combination AG/AG can be fully explained by the presence of the A in the AG of OCA2’.

	Primary analysis (combined 100KG_P and Bordeaux albinism cohort)	Replication analysis (UK Biobank)	Secondary analysis (100KG_P)	Additional analysis following comment #1.3 (Bordeaux albinism cohort)
	case-control odds ratio (95% confidence interval)			unsolved-solved odds ratio ^{(95%} confidence interval)
Group C GG(TYR)/AG(OCA2) [OCA2 heterozygotes]	4.6 ^(2.0 – 9.2)	8.6 ^(0.9 – 37.3)	2.5 ^(0-19.3)	1.0 ^(0.42-2.26)
Group D AG (TYR)/AG(OCA2) [dual heterozygotes]	14.0 ^(6.8 – 26.5)	16.3 ^(3.1-57.3)	14.8 ^(1.5-71.9)	3.3 ^(1.39-7.7)

Selected data from Supplementary Tables 2, 3, 6 and 7 are shown.
100K_GP corresponds to Genomics England 100,000 Genomes Project.

The above Table highlights that the effect (odds ratio) of dual heterozygosity for TYR c.1205G>A and OCA2 c.1327G>A (group D) appears to be >2 times higher than that of heterozygosity for OCA2 c.1327G>A alone (group C) (see also response #1.3 and Supplementary Table 7).

We have now added a mention to the caveat highlighted by the reviewer in the penultimate paragraph of the manuscript (see response #1.8) and present relevant data/comparisons in a new supplementary table (Supplementary Table 7).

Comment #2.3: An important point to consider is that the OCA2 c.1327G>A variant, when present in a homozygous manner, acts as a fully penetrant causal allele, whereas the TYR c.1205G>A variant clearly is a hypomorphic allele. Is it thus logical to test an hypothesis that they have ‘equal weight’ effects, when both present in a heterozygous manner? Can the authors elaborate on this?

Response #2.3: As the reviewer mentions, the OCA2 c.1327G>A variant and the TYR c.1205G>A variant do not have equal weight effects. We show that this is not the case.

Notably, the key hypothesis of our study is that dual heterozygosity for these two variants is associated with a significantly increased probability of receiving a diagnosis of albinism. The

combined effect may be additive or non-additive but the variants certainly do not need to have equal weights. We have included a comment on this in the footer of Supplementary Table 2:

“It is noted that the TYR:c.1205G>A and OCA2:c.1327G>A variants appear to have unequal effects. This is unsurprising given the fact that the common TYR:c.1205G>A change is considered to be a ‘hypomorphic’ variant associated with relatively mild forms of albinism.

Comment #2.4: *A synergistic effect would mean that small, subclinical effects of the alleles on their own, when present together, give rise to an effect that is bigger than the effects of each allele separately. Based on previous studies, there does not seem to be a direct interaction between the TYR and OCA2 proteins. Irrespective of an additive or synergistic mechanism of disease, would it not make sense that albinism cases carrying TYR c.1205G>A and another causal TYR variant in trans, would have a more severe phenotype (i.e. severe endophenotypes, such as retinal thickness and central visual acuity) if the OCA2 c.1327G>A is present? Is there evidence for this? As the latter is quite rare, this may be difficult to study. But the converse also could be true. If an albinism case carries the OCA2 c.1327G>A variant with another OCA2 variant in trans, the presence of the TYR c.1205G>A could also give rise to a more severe phenotype. This kind of data could be presented in Supplementary Table 4, but unfortunately, the data for groups B and D were lumped together. The underlying endophenotype data for the subgroups should be provided.*

Response #2.4: As mentioned in our response to comments #1.5 and #2.1, we have now split the mixed genotype groups and re-analysed the data. Furthermore, clinical information is now included for selected subgroups in Supplementary Data 3 (see also responses to comments #1.6 and #1.7).

The reviewer highlights that a cumulative burden of variants (in TYR, OCA2 and, potentially, other albinism-related genes) may affect the severity of the phenotype in individuals with albinism. While this is plausible, it is challenging to robustly prove this hypothesis and this is not the focus of our study. Here, we wanted to inform molecular diagnostics by studying if heterozygosity for TYR:c.1205G>A (p.Arg402Gln) [rs1126809] and OCA2:c.1327G>A (p.Val443Ile) [rs74653330] could lead to an increased probability of receiving a diagnosis of albinism. We sought to understand if this dual heterozygous genotype has a sufficiently significant baseline effect on melanin synthesis (in developing retinal pigment epithelia and melanocytes) to cause notable clinical signs/symptoms.

With regards to the interaction between the TYR and OCA2 proteins: although the reviewer correctly highlights that there is no classic direct protein-protein interaction between them, numerous previous studies have pointed to a functional interaction between these two molecules. It is now widely accepted that OCA2 has a role in the pH regulation of the melanosome which, in turn, can affect the catalytic activity of TYR (i.e. tyrosinase, the rate-limiting enzyme for the synthesis of melanin) (Pavan

& Sturm, *Annu Rev Genomics* 2019 [DOI: 10.1146/annurev-genom-083118-015230]). A mention to this can be found in P7-8, L119-124.

Minor points:

Comment #2.5: *In literature there is some controversy over the causal nature of the TYR c.1205G>A variant, but it very likely acts as a hypomorphic allele when together with a severe TYR allele. The high allele frequency of TYR c.1205G>A in European populations is striking – is reduced penetrance observed in individuals with TYR c.1205G>A and a second allele? The authors could provide further discussion that the TYR c.1205G>A variant may be of low penetrance and the high allele frequency observed could have a cumulative effect on the overall risk of albinism.*

Response #2.5: We have now added four mentions to this

- Fig.1 legend (P7): “TYR c.-301C>T [rs4547091] and TYR c.575C>A (p.Ser192Tyr) [rs1042602] (...) have been previously shown to modify the effect of TYR:c.1205G>A. This common missense change appears to act as a ‘hypomorphic’ variant.¹⁸⁻²⁰”
- in P8, L133-136: “The particularly common TYR:c.1205G>A change has been associated with incomplete penetrance (and variable expressivity) and previous studies have pointed to the potential for other TYR variants to modify its effect.^{18-20,24,33-35}”
- in P18, L351-353: “The effect of one of the studied changes (TYR c.1205G>A) has been previously shown to be modified by the following two variants which have been added as a covariate to the regression model (...)”
- Supplementary Table 2 footer: “It is noted that the TYR:c.1205G>A and OCA2:c.1327G>A variants appear to have unequal effects. This is unsurprising given the fact that the common TYR:c.1205G>A change is considered a “hypomorphic” variant associated with relatively mild forms of albinism.”

Comment #2.6: *The introduction could benefit from some additional information into the previous reported variants implicated in albinism (from line 89) to introduce the frequent genes and variants that are reported in the methods section. E.g. line 237: c.-301 TYR variant is mentioned, but it is not apparent until line 316 the relevance of this. Likewise, c.575C>A/p.Ser192Tyr was found more often in cis with c.1205G>A in compound heterozygous cases, than seen in the general population. Although there does not seem to be an added effect of c.575C>A on the phenotype in its presence, the authors may mention the existence of this frequent variant.*

Response #2.6: As mentioned in our response to the previous comment (comment #2.5), we have now highlighted the potential for TYR c.-301C>T [rs4547091] and TYR c.575C>A (p.Ser192Tyr) to modify the effect of of TYR:c.1205G>A in: P7 (Fig.1 legend); P8, L133-136; P18, L351-353; and the footer of Supplementary Table 2.

Comment #2.7: *The authors may also mention the Burkard et al 2018 CNGA3/CNGB3 digenic triallelic inheritance.*

Response #2.7: We have now included the Burkard *et al* 2018 reference in P4, L70. We have however chosen not to extensively discuss this key paper as it describes a different oligogenic pattern (involving two recessive genes and the combination of biallelic alterations in one gene and a single heterozygous variant in the other).

Comment #2.8: *Supplementary data 1: Nomenclature of deletions is not according to HGVS*

Response #2.8: We have now made relevant changes in Supplementary Data 1.

Comment #2.9: *Why are synonymous variant such as HPS5:c.219G>A (p.Arg73=) and SLC45A2:c.1518C>T (p.Val506=) provided? What is the evidence that they are causal?*

Response #2.9: The following ACMG-AMP (American College of Medical Genetics and Genomics and the Association for Molecular Pathology) 2015 criteria were applied to these variants resulting in them being classified as likely pathogenic (ACMG-AMP class 4):

- *HPS5:c.219G>A* (p.Arg73=) [detected in 1 study participant]
 - PP3 : 1 bp from donor splice site, predicted to induce splice defect and loss-of-function
 - PM2 : absent from gnomAD
 - PM3 : detected *in trans* with another pathogenic *HPS5* variant
- *SLC45A2:c.1518C>T* (p.Val506=) [detected in 5 study participants]
 - PM2 : rare with no homozygous in gnomAD
 - PP5 : likely pathogenic in Clinvar
 - PM3 : detected *in trans* with another *SLC45A2* pathogenic variant
 - PS4 : prevalence is higher in cases than controls

Comment #2.10: *Please check all tables and figures to capitalize the headings for consistency (e.g. in Supplementary Data 1: 'rare single nucleotide...' should be 'Rare single nucleotide...', and 'male'/'female' in column B should be capitalized, etc.*

Response #2.10: We have now made the suggested changes in the headings of Supplementary Data 1 and 2.

Comment #2.11: *Italicize gene names in Supplementary Data 1 and 2 tables*

Response #2.11: We have now made the suggested changes in the gene names included in Supplementary Data 1 and 2.

Comment #2.12: *Line 130: the minor allele frequencies for both variants are reported. This sentence could be better linked to the previous sentence.*

Response #2.12: We have now re-phrased and expanded the relevant section in P8, L130-136.

REVIEWER 3 COMMENTS

This paper by Green and al. investigates the epistatic effect of two mutations already shown to have a main effect on albinism. These mutations are in two genes which were possibly interacting, and thus there is a strong prior for possible interaction.

Case control design is comparing albinism cases mainly of French origin and controls from Britain. While they are both from European descent, we could expect differences that could lead to false positives – maybe even more for rare variants. The authors try to account for this design. Interestingly, this study finds a very strong association of double heterozygotes with albinism condition in this heterogeneous case controls design and some association with endophenotypes within the homogenous British population, which is a bit reassuring. One of the main questions I have is how much of this association is due to main effects rather than to pure epistasis.

Comment #3.1: *1/ I am not sure what the difference is between qualitative and quantitative approach. Testing interaction is estimated quantitatively (interactions test, likelihood models ...). While not crucial, I do not see why the approach used by the authors is declared quantitative as opposed to qualitative ones.*

Response #3.1: To minimise confusion, we have removed all references to the quantitative/qualitative nature of the different approaches that we used.

Comment #3.2: *2/ I am not sure what is the difference, in essence, between the two analyses. The first one is a logistic regression while the other one is comparing various groups of (di-genotypes). Analysis 2 can fit into logistic regression so this is just an unbalanced way to represent the analyses. Maybe better say that analysis one is « linear » in the sense that they are testing association of the number of rare alleles and their product (if I understand well what they call logistic regression).*

Response #3.2: In response to feedback by all three reviewers (including comments #1.2, #2.1, #2.4, #3.1, #3.2 and #3.5) we have now modified and clarified the analytical approach.

Comment #3.3: *3/ Line 212 : I find the approach description lacks a bit of clarity. What are targeted analyses? It is clarified later on, but explaining in the main text how this could help correcting for possible stratification.*

Response #3.3: We have now expanded the relevant section (P13, L205) which now reads:

“Subsequently, we performed a targeted secondary analysis using only Genomics England 100,000 Genomes Project data which allowed us to study case and control groups of high genetic similarity. The results supported our key findings and increased confidence in the validity of the detected associations (Supplementary Fig.1b and Supplementary Table 6).”

Comment #3.4: 4/ Why are you only using 35 « supposedly neutral » to estimate the inflation factor. This inflation factor 1.04 seems a bit low for a study comparing cases and controls from two countries (high prior of false positive due to stratification) : this is a strong assumption, which may be true but really needs strong support, with as much SNPs as possible.

Response #3.4: We agree with the reviewer that as many SNPs as possible should be used to calculate the inflation factor (λ GC). A key issue however is that, for λ GC calculation, the utilised SNPs should be unlinked, random and not associated with the studied phenotype/trait.

For the purpose of this study, we had to identify a set of suitable SNPs that were analysed both by the Bordeaux gene panels and the 100K_GP genome sequencing assays. Notably, the Bordeaux panels were designed to test pigmentation-associated loci and selecting truly neutral SNPs for λ GC analysis among the variants genotyped by the gene panel required careful consideration. We chose to focus on variants with a CADD score less than 5 and identified a set of 35 SNPs (Supplementary Table 5). We then used Firth regression analysis to estimate λ GC and found it to be 1.04 (which is close to the expected λ GC value of 1).

We note that the cohorts and λ GC-related methodology used here are the same to those in a recent paper published by our group in Nature Communications (Michaud et al. Nat Comms 2022 [DOI: 10.1038/s41467-022-31392-3]).

Further information on this can be found in our response to the related comment #3.5 below.

Comment #3.5: 5/ I am not convinced that the genetic stratification is corrected for and propose to check the allele frequency of both the genetic variants and combination of genotypes in a French reference population.

Response #3.5: We agree with the reviewer that correcting for population stratification is key in case-control genetic studies. We tried to access French 'control' population cohorts but it became clear that this cannot be done in a reasonable timeframe. It is noted that the response from one of the leads of the French Exome Project (FREX) highlighted that access to the relevant exome data for the purposes of our project would not be possible

As an alternative (and following a recommendation by reviewer #1 - comment #1.3) we used the 'solved' subset of the Bordeaux albinism cohort as a surrogate 'control' cohort. This additional analysis supported our conclusions (Supplementary Table 7).

Overall, we discuss the potential implications of imperfect genetic stratification as a caveat in the penultimate paragraph of the manuscript and have performed a number of additional analyses to

increase confidence on the validity of our results (see also response #1.8). More broadly, we believe that our findings are valid not only on the basis of the detected median λ GC in the primary analysis but also because of the results of secondary analyses conducted in targeted subgroups in which there was significant genetic similarity between cases and controls such as:

- (i) the *Genomics England 100,000 Genomes project* (Supplementary Fig.2b; Supplementary Table 6);
- (ii) the *UK Biobank* (Fig.3a; Supplementary Table 3);
- (iii) the *Bordeaux albinism cohort* (as per comment #1.3) (Supplementary Table 7).

Comment #3.6: *6/ It is not clear how the interaction model is tested. The authors claim that there is interaction but it tests alleles which are already found to increase the risk themselves. Again, please clarify the exact model which is being tested - for instance tease apart the beta and p-value for main and interaction effects. I well noticed that the increase in OR on Figure 2 seems non linear, but we'd need a more formal testing as the sample size is small.*

Response #3.6: In response to feedback by all three reviewers (including comments #1.2, #2.1, #2.4, #3.1, #3.2 and #3.5) we have now modified, simplified and clarified the analytical approach. We are no longer lumping genotype groups and it is now clearer what is the expected impact of the individual alleles tested (see also responses #2.1 and #2.2). With regards to the presence of an additive or non-additive interaction effect: we chose not to speculate on this given the sample size limitations. The following comment has now been included to the footer of Supplementary Table 2:

"Although the above observations can be viewed as suggestive of a non-additive interaction between TYR:c.1205G>A and OCA2:c.1327G>A, such speculation would be inappropriate in the absence of focused analyses in studies with larger sample sizes."

We hope that these changes have addressed the reviewers' comments and that it will now be possible to consider our manuscript as suitable for publication in *Nature Communications*.

Thank you for your help and time,

Yours faithfully,

Panos Sergouniotis, David Green, Benoit Arveiler and co-authors

REVIEWER COMMENTS

Reviewer #2 (Remarks to the Author):

The authors addressed all my points very well.

In the Methods, while listing many genes, put them always in alphabetic order.

Reviewer #3 (Remarks to the Author):

The authors made everything they could to answer to and address my concerns (shared by the other reviewers).

I appreciate how it was done and understand that it is very difficult for the time being to have access to additional French data right now, I still have some concerns.

- replication in UKBiobank seems to rely on a really limited number of patients (24).

- I feel better how the lambda GC is using so few SNPs. I think the authors should provide an "association" test for the same 4 groups in cases from Bordeaux vs cases of British European descent.

- if we were convinced that there is no inflation due to demographic stratification, it is still not easy to ease apart an effect due to single A allele in OCA as compared to a double heterozygote effect. This is especially true in endophenotypes. I noticed efforts to answer reviewers 2 similar concern but feel this should be made more clear./discussed.

All in all, there are convincing results related to a dose effect risk increase with some questions related to a main OCA2 c.1327G>A effect alone.

Reviewer #4 (Remarks to the Author):

The authors have addressed some of the comments, however the most significant concerns related to study design and case-control selection still remains.

Comment related to 1.1 This maybe an underlying issue with retrospective heterogeneous studies. The case selection based on a loose definition of albinism (nystagmus or foveal hypoplasia) is problematic as it could introduce other differentials linked to nystagmus or foveal hypoplasia within the case definition. There are specific diagnostic criteria proposed for albinism (PMID: 30098354) which should be followed. In the control cohort since the level of phenotyping would vary, it would not be possible to say with certainty whether the same traits used to define the albinism case cohort is present. Based on reply to comment 1.7 the authors acknowledge that the chosen traits to define albinism in their study will likely be present in the control cohort. This is problematic as it introduces bias in the way the cases and controls are defined and does not follow uniform standards for case and control definitions.

The variant inclusion/exclusion criteria also present some confusion and certain aspects related to this is unclear. The authors exclude all pathogenic or likely pathogenic variants and only include unsolved cases (figure 1). However, it is unclear whether the cis-YQ haplotype was classified as pathogenic or not. Based on recent work it is convincing that the cis-YQ is a pathogenic haplotype (PMID: 30679655, 35803923, 37327787) and represents approximately 19.1% of TYR pathogenic alleles (PMID: 37327787). Therefore, this haplotype should be treated as pathogenic. The Bordeaux cohort (n=1208)

has previously been presented including the linked genotypes to solved/unsolved cases (PMID: 35803923). In the current manuscript the authors are presenting a subset of this (n=1015, note that supp data 1 says the cohort size is 1016). Among unsolved cases (as per their previous study, PMID: 35803923) there are 18 individuals that have the OCA2 variant c.1327G>A. Further breakdown based on TYR c.1205G>A genotype (GA is present in 11 and GG in 7). Thus within the Bordeaux cohort the number of cases of dual heterozygosity is 11 (or 10 as per supp data 1), if one takes into account cis-YQ haplotype this leaves around 3 cases with dual heterozygosity based on the published Bordeaux cohort (PMID: 35803923). These are quite small numbers to infer population based genetic risk studies.

Was the sequencing methodology/genotyping the same for all participants within the Bordeaux cohort? Among some of the unsolved cases, a few only had <8-gene-sanger sequencing, or 12 gene panel. Therefore it is possible that some of the unsolved cases could have pathogenic variants within known albinism genes that were not sequenced. Further clarification is also required whether all of these participants had CNV analysis (see comment 1.2 below).

Within the 100K_GP cohort, the proportion of cases that meet the inclusion criteria seems to be nearly double (~30%) compared to Bordeaux cohort (~17%). One would expect within the same ancestry groups that these number should be similar.

Comment related to 1.2 – thank you for including CNV analysis as part of the variant pathogenicity workup. The authors have mentioned that this has strengthened the detected signals. Presumably this has also further decreased the number of unsolved cases. However in the revised manuscript it is unclear how many were excluded as a result of this. Based on revised figure 1, only 1 case seems to be excluded.

Comment related to 1.5 – it would be helpful to have the numbers within the figure to readily visualise them.

1.6 – The endophenotypes of visual acuity and central retinal thickness are influenced by many other factors such as refractive error, media opacities, acquired (known and unknown) retinal disorders, genotype (including carriers). It is also well known that carriers of albinism can have slightly reduced visual acuity and increased central retinal thickness. Have the authors excluded possible albinism carriers (including the pathogenic cis-YQ haplotype) and similarly accounted for other causes of reduced vision when constructing this cohort from UK Biobank. These details are missing. Once again numbers with a possible flow chart would be useful. Similarly please include numbers within supplementary table 4. Having a validation/replication cohort is important, however the case-control comparisons from the UK Biobank are extremely small numbers to draw meaningful conclusions and validity for a replication cohort.

1.7 – Since the numbers are small, performing segregation analysis of these variants within pedigrees would provide more substantial evidence whether different albinism traits are truly related to these variants. As mentioned previously a loose definition of albinism based on possible presence of one of two traits is not a robust case definition and the authors should aim to highlight which of the phenotypic spectrum is linked to dual heterozygosity.

Re: Second revision of NCOMMS-22-51271 (“The co-occurrence of genetic variants in the TYR and OCA2 genes confers susceptibility to albinism”)

Many thanks for reviewing our manuscript and for giving us the opportunity to submit an appeal and a revised version. Please find below a point-by-point response:

REVIEWER 2 COMMENTS

Comment #2.1R: *The authors addressed all my points very well.*

In the Methods, while listing many genes, put them always in alphabetic order.

Response #2.1R: We would like to thank reviewer 2 for their helpful and insightful comments which have allowed us to improve our manuscript. We have now made the recommended modification in the Methods (page 14, line 291)

REVIEWER 3 COMMENTS

The authors made everything they could to answer to and address my concerns (shared by the others reviewers). I appreciate how it was done and understand that it is very difficult for the time being to have access to additional French data right now, I still have some concerns.

Comment #3.1R: *- replication in UKBiobank seems to rely on a really limited number of patients (24).*

Response #3.1R: It has now been possible to obtain access to the Primary Care (GP) data for UK Biobank (UKB) participants (resource 591). Including these data has led to an increase in the overall number of eligible UKB cases from 24 to 39. We have now re-run the UK Biobank analyses (also incorporating suggestions from reviewer #4 - see response #4.9R) and describe the analytical approach in Suppl. Fig.2 and the findings in Suppl. Tables 3-6.

More broadly, it is worth noting that in addition to the primary analysis (in which we used data from the Bordeaux albinism cohort and the Genomics England 100,000 Genomes (100K_GP) dataset; Fig.2 and Suppl. Fig.1a), we have performed two secondary analyses:

- one in 100K_GP alone (Suppl. Fig1b) and
- one the UKB. (Fig.3)

The UKB work focused both on repeating and expanding our study and involved three additional sub-analyses:

- (i) one comparing cases with controls (Fig.3a)
- (ii) one comparing visual acuity in individuals with specific genotypes (Fig.3b)
- (iii) one comparing retinal thickness in individuals with specific genotypes (Fig.3c)

As the reviewer highlights, the number of cases that were used in sub-analysis (i) was small (n=24 initially, n=39 after incorporating the Primary Care records). Nonetheless, it is encouraging that, even with small numbers, a statistically significant result could be obtained.

We agree with the reviewer that the UKB findings, even with the additional cases, on their own, should be approached with caution. However, the value of these data as supporting/triangulating evidence is notable in the context of [a] observations in the Bordeaux and the 100K_GP cohorts and [b] sub-analyses (ii) and (iii) in UKB. It is also worth highlighting that the signal was retained after substantially altering the analytical approach in response to the reviewers' comments.

Comment #3.2R: - *I feel better how the lambda GC is using so few SNPs. I think the authors should provide an "association" test for the same 4 groups in cases from Bordeaux vs cases of British European descent.*

Response #3.2R: In response to this comment, we performed an additional Bordeaux cases vs. 100K_GP cases analysis. This involved applying the same Firth logistic regression model used in the cases vs. controls studies. We only focused on cases presumed to have European-like ancestries. Reassuringly, no statistically significant differences could be detected in this case-to-case comparison; the results are shown in the table below.

genotype			odds ratio ^b	95% confidence interval	unadjusted p-value	number of cases (Bordeaux / GeL)
TYR: c.1205G>A (p.Arg402Gln) [rs1126809]	OCA2: c.1327G>A (p.Val443Ile) [rs74653330]	group				
GG	GG	A	0.7	0.3 – 1.5	0.3	12 / 91
AG	GG	B	1.8	0.8 – 3.9	0.14	15 / 56
GG	AG	C	0.4	0 – 3.3	0.5	0 / 7
AG	AG	D	0.8	0.6 – 1.1	0.8	1 / 9

Comment #3.3R: - *if we were convinced that there is no inflation due to demographic stratification, it is still not be easy to tease apart an effect due to single A allele in OCA as compared to a double*

heterozygote effect. This is especially true in endophenotypes. I noticed efforts to answer reviewers 2 similar concern but feel this should be made more clear/discussed.

Response #3.3R: We have now added a further mention to this caveat in the penultimate paragraph of the manuscript (page 13, line 244-255):

“Another potential caveat is that the detected signal in dual heterozygotes for TYR:c.1205G>A and OCA2:c.1327G>A could, to a degree, be driven by undetected pathogenic variation in either TYR or OCA2. This cannot be excluded despite the fact that medical-grade genetic analyses were undertaken. It can be argued however that significant confounding is unlikely especially given that the effect of dual heterozygosity for TYR:c.1205G>A and OCA2:c.1327G>A appears to be consistently greater than that of genotypes involving heterozygosity for either TYR:c.1205G>A or OCA2:c.1327G>A alone (Supplementary Tables 2, 3, 8 and 9). Nonetheless, statistically significant results were also obtained for individuals carrying the OCA2:c.1327G>A change in heterozygous state (without TYR:c.1205G>A) and future studies will provide further insights into the extent to which this change alone drives the observed signal.”

Comment #3.4R: *All in all, there are convincing results related to a dose effect risk increase with some questions related to a main OCA2 c.1327G>A effect alone.*

Response #3.4R: We thank the reviewer for the helpful and informative feedback. In this paper, in addition to dual heterozygotes, we present data on individuals who are heterozygous for the OCA2 c.1327G>A variant alone (group C). We did not want to over-interpret the results for this genotype group as it was not the primary focus of the study (especially given the lack of a significant p-value in the Suppl. Fig.1b and Suppl. Table 9 analyses). We appreciate the reviewer's conclusion that the results are convincing.

REVIEWER 4 COMMENTS

Comment #4.1R: *Comment related to 1.1 This maybe an underlying issue with retrospective heterogeneous studies. The case selection based on a loose definition of albinism (nystagmus or foveal hypoplasia) is problematic as it could introduce other differentials linked to nystagmus or foveal hypoplasia within the case definition. There are specific diagnostic criteria proposed for albinism (PMID: 30098354) which should be followed.*

Response #4.1R: We would like to thank the reviewer for giving us the opportunity to clarify the case selection criteria. We feel that there are misperceptions about how we defined albinism. Individuals with just foveal hypoplasia or just nystagmus were not considered cases.

The approach that we used was similar to that utilised in a recent related publication from our group: *Nat Commun* 13, 3939 (2022) [10.1038/s41467-022-31392-3]. In brief:

- for the cases in the Bordeaux cohort: independent assessment of the available phenotypic information was undertaken by two clinicians with extensive experience in the field. If insufficient clinical information was available, further data were requested/obtained. As mentioned in the manuscript (page 14, lines 280-285), "*Information on the dermatological and ophthalmological phenotypes was available for all affected individuals. The relevant clinical data were assessed independently by two clinicians with experience in the field of albinism and only cases with a consensus clinical impression were included. It is noted that and each of these cases had at least one of the key ocular features of albinism, i.e. infantile nystagmus or absence of a foveal pit (prominent foveal hypoplasia) (Supplementary Table 10).*" It is worth highlighting again though that presence of either of these ocular phenotypes in isolation was not considered sufficient.
- for the cases in the Genomics England dataset the ICD-10 term "Albinism" [E70.3] and/or the HPO terms "Albinism" [HP:0001022], "Partial albinism" [HP:0007443] or "Ocular albinism" [HP:0001107] were assigned by the recruiting specialist (as discussed in page 16, lines 350-352).
- for the cases in the UK Biobank: the ICD-10 term "Albinism" [E70.3] has been recorded in data field 41270 (Diagnoses - ICD10) or in resource 591 (Primary Care record data) (as discussed in page 19, lines 455-457).

The reviewer mentions specific diagnostic criteria proposed for albinism and references a publication by Kruijt *et al.* from 2018 (PMID: 30098354). While this is an important paper for the field, the proposed criteria are challenging to implement (especially in young children in which comprehensive ophthalmic phenotyping is not possible) and are far from being universally accepted. Some of the criticisms to this rigid approach are discussed in a paper from Dumitrescu *et al.* 2021 (PMID: 34251969). A key statement from this University of Iowa publication is: "*the Kruijt et al. criteria for diagnosis, when used without genetic testing may miss up to 25% of albinism cases*". This leads to a second important point: The reviewer states that the Kruijt *et al.* criteria should be followed. However, genetic testing is a key component of these criteria making them unsuitable in the context of case-control studies focusing on improving genetic diagnostics.

It is worth highlighting that we are involved in active discussions within the ClinGen International Albinism Expert Panel (which includes >20 experts from 10 countries) on how to clinically define an 'albinism case'. Two of us (PIS and BA) are members of this group and one is a co-chair. The current consensus is that the diagnosis of albinism is based on a 'gestalt' (rather than specific diagnostic criteria) and our approach is in line with this.

Overall, in response to this comment, we have provided further detail on how cases were defined in the Bordeaux cohort (e.g. in page 14, line 280-285) and have added the following Table which summarises the case selection criteria (Suppl. Table 10).

Supplementary Table 10. Albinism case definition in the different cohorts used in this study

cohort	number of cases	approach/criteria
University Hospital of Bordeaux Molecular Genetics Laboratory cohort	1,015	 identified through the database of the University Hospital of Bordeaux Molecular Genetics Laboratory, France. independent assessment of available phenotypic data was performed by two clinicians with experience in albinism diagnostics; only cases with a consensus clinical impression of albinism were included ^a.
Genomics England 100,000 Genomes Project cohort	105	 enrolled across multiple clinical specialties in designated Genomic Medicine Centres (GMC) within the English National Health Service (NHS); standardised baseline clinical data were recorded with the use of Human Phenotype Ontology (HPO) terms guided by disease-specific data models. the primary clinical data were inspected; individuals with one of the following codes were included in the relevant case group:  the ICD-10 term "Albinism" [E70.3] the HPO term "Albinism" [HP:0001022] the HPO term "Partial albinism" [HP:0007443] the HPO term "Ocular albinism" [HP:0001107]
UK Biobank cohort	39	 recruited in dedicated UK Biobank assessment centres; the baseline visit included a touchscreen questionnaire, physical measures and biological sampling; crucially, a range of data from other national datasets were incorporated including primary care, screening programmes, and disease-specific registries. a summary of the distinct ICD-10 diagnosis codes that each participant has had recorded across all their hospital inpatient records in either the primary or secondary position was available in data field 41270. primary care data recorded by health professionals working at general practices was available in resource 591 (category 3000). UK Biobank participants with the ICD-10 term "Albinism" [E70.3] in data field 41270 or resource 591 were included in the relevant case group.

^a The principle that underlies this process is that the diagnosis of albinism, like many other diagnoses in medicine, is based on a gestalt, a constellation of observations and findings. There is no single sign or symptom that is perfectly diagnostic and clarity can only be gained when different pieces of available information (clinical, genetic, imaging, electrophysiological etc) are jointly inspected and combined using Bayesian reasoning.

Comment #4.2R: *In the control cohort since the level of phenotyping would vary, it would not be possible to say with certainty whether the same traits used to define the albinism case cohort is present. Based on reply to comment 1.7 the authors acknowledge that the chosen traits to define albinism in their study will likely be present in the control cohort. This is problematic as it introduces bias in the way the cases and controls are defined and does not follow uniform standards for case and control definitions.*

Response #4.2R: Most individuals in the control cohort were recruited by clinicians (typically geneticists) in dedicated Genomic Medicine Centres (GMC) within the English National Health Service. While we agree with the reviewer that the level of phenotyping in the 29,451 controls is

likely to vary (and to be, potentially, on average, less comprehensive compared to that in the 1,120 cases), it is worth highlighting that this situation is not uncommon in large-scale case-control genetic studies and it is generally accepted that the risk of bias in this context is small.

To gain insights into this, we applied an iterative approach by randomly assigning 10 controls at a time to the case group to simulate misdiagnosis. We then ran the Firth logistic regression model as described in the manuscript for each iteration to study the impact on the odds ratio for genotype D. Below, we show the change in the OR for genotype D as 10 misdiagnosed controls are iteratively added to the case group. We focus on the results for the first 10 iterations of this process. At the 10th iteration (representing 100 missed albinism diagnoses), the point estimate of the OR remains high at 7.4 with a p-value of 2.6×10^{-6} . We hope that these observations provide reassurance that misdiagnosis within the control group is not a likely explanation for our findings.

The reviewer mentions that in “comment 1.7 the authors acknowledge that the chosen traits to define albinism in their study will likely be present in the control cohort”. It is unclear to us to which part of our response to comment 1.7 this statement relates to.

The reviewer also mentions that the way the cases and controls are defined introduces bias as we did “not follow uniform standards for case and control definitions”. We respectfully disagree with this comment and would like to highlight that identical processes were followed in:

- the 100K_GP analysis (Suppl. Table 8),

- the UK Biobank analysis (Suppl. Tables 3-6)
- the solved versus unsolved analysis in the Bordeaux cohort (Suppl. Table 9).

Overall, we believe that our approach to case and control selection is robust and that the above criticisms are due to misperceptions around the albinism case selection criteria that we use (see also response #4.1R). This stems from the lack of a detailed description of these criteria in the previous version of the manuscript. We hope that Suppl. Table 10 and the additional text discussed in response #4.1R provide reassurance/clarity.

Comment #4.3R: *The variant inclusion/exclusion criteria also present some confusion and certain aspects related to this is unclear. The authors exclude all pathogenic or likely pathogenic variants and only include unsolved cases (figure 1). However, it is unclear whether the cis-YQ haplotype was classified as pathogenic or not. Based on recent work it is convincing that the cis-YQ is a pathogenic haplotype (PMID: 30679655, 35803923, 37327787) and represents approximately 19.1% of TYR pathogenic alleles (PMID: 37327787). Therefore, this haplotype should be treated as pathogenic.*

Response #4.3R: We agree with the reviewer that this haplotype should be treated as pathogenic. Indeed, we have classified it as such. In the Methods section (page 17, line 386-400) we state that we excluded individuals who carried genotypes consistent with a molecular diagnosis of albinism including study subjects with

- the *TYR* c.[-301C;575C>A;1205G>A] or *TYR* c.[-301C;575C;1205G>A] albinism-related haplotype in homozygous state
- the *TYR* c.[-301C;575C>A;1205G>A] or *TYR* c.[-301C;575C;1205G>A] albinism-related haplotype in heterozygous state plus a heterozygous pathogenic or likely pathogenic variant in *TYR*.

This might have not been immediately apparent as we use the HGVS nomenclature term for the cis-YQ allele: *TYR* c.[-301C;575C>A;1205G>A]

Comment #4.4R: *The Bordeaux cohort (n=1208) has previously been presented including the linked genotypes to solved/unsolved cases (PMID: 35803923). In the current manuscript the authors are presenting a subset of this (n=1015, note that supp data 1 says the cohort size is 1016). Among unsolved cases (as per their previous study, PMID: 35803923) there are 18 individuals that have the *OCA2* variant c.1327G>A. Further breakdown based on *TYR* c.1205G>A genotype (GA is present in 11 and GG in 7). Thus within the Bordeaux cohort the number of cases of dual heterozygosity is 11 (or 10 as per supp data 1), if one takes into account cis-YQ haplotype this leaves around 3 cases with dual heterozygosity based on the published Bordeaux cohort (PMID: 35803923). These are quite small numbers to infer population based genetic risk studies.*

Response #4.4R: We thank the reviewer for the opportunity to provide more clarity:

- the starting point for our analysis in the Bordeaux cohort was 1,015 cases (apologies for the typo in Suppl. Data 1; this has been corrected). This is smaller than the 1,208 cases included in our previous *Nature Communications* paper (PMID: 35803923). This is because we removed 193 individuals for which the position corresponding to the *TYR* c.-301C>T variant has not been genotyped (“NA” in column H of the Suppl. Dataset 1 in PMID: 35803923). A mention to this can be found in the Methods (page 14, lines 285-286): “*No pre-screening based on genotype was undertaken other than selecting individuals in whom position c.-301 of TYR was sequenced.*” We felt that having incomplete data for this key covariate could potentially lead to spurious results and preferred to focus on the subset for which comprehensive genetic information was available.
- as mentioned in the reviewer’s comment and shown in Suppl. Data 2, the cases of dual heterozygosity in the primary analysis were 10+1. As we note in the Methods (page 14, lines 286-287): “*Only individuals who were not knowingly related were included*”. We were recently made aware that two probands in the genotype group D (dual heterozygosity) are related and we therefore removed the case that presented second. The number of cases in group D (dual heterozygotes) had to be amended from 11 to 10. This led to a change in the odds ratio:
 - from: odds ratio 14.0; 95% confidence interval 6.8 – 26.5; p-value 1.9×10^{-9}
 - to: odds ratio 12.8; 95% confidence interval 6.0 – 24.7; p-value 2.1×10^{-8}
- it is highlighted that we have taken into account the cis-YQ haplotype and have included the *TYR*:c.575C>A and *TYR*:c.-301= alleles as covariates to our analyses. The reviewer eludes to the fact that only 3 cases had the *TYR*:c.1205G>A and *OCA2*:c.1327G>A variants in heterozygous state AND did not carry the common *TYR* c.575C>A change (Suppl. Data 2). Our impression is that the reviewer speculates that the effect of dual heterozygosity for *TYR*:c.1205G>A and *OCA2*:c.1327G>A can be modified by the presence of *TYR* c.575C>A. Indeed such modification is possible (as we mention in the Discussion, p.12, lines 222-229) and the confidence interval of the corresponding odds ratio reflects some of this variability. Clearly however, our conclusions are not based on findings in 3 individuals. We have now included the following (Suppl. Table 9): “*It can be speculated that the expressivity/penetrance of dual heterozygosity for TYR:c.1205G>A and OCA2:c.1327G>A can be modified by other non-genetic or genetic factors (such as the presence of TYR:c.575C>A (p.Ser192Tyr)); studies in large, comprehensively phenotyped cohorts are expected to provide further insights into more complex variant interaction patterns.*”
- the reviewer comments that “*these are quite small numbers to infer population based genetic risk studies*”. It is noted that our intention is not to infer an absolute population-based risk (e.g.

in the context of phenotype-agnostic newborn screening). Instead, we wanted to obtain an estimate that could assist within a Bayesian framework that combines clinical and genetic data (similar to a concept described in a publication linked to the American College of Medical Genetics and Genomics [ACMG] (PMID: 29300386). As the reviewer highlights, the number of individuals in one of the four studied groups in one of the multiple analyses that we conducted is 10. It is unclear to us what is a sufficiently large number (for each subgroup) in this context and we feel that it would be inappropriate to ignore statistically significant observations that replicate in different cohorts/analyses. We agree however with the reviewer about the importance of carefully considering potential sources of bias, especially when small case numbers are available, and we appreciate their useful suggestions that have allowed us to improve our analytical approach (e.g. in UKB, see comment #4.9R)

Comment #4.5R: *Was the sequencing methodology/genotyping the same for all participants within the Bordeaux cohort? Among some of the unsolved cases, a few only had <8-gene-sanger sequencing, or 12 gene panel. Therefore it is possible that some of the unsolved cases could have pathogenic variants within known albinism genes that were not sequenced. Further clarification is also required whether all of these participants had CNV analysis (see comment 1.2 below).*

Response #4.5R: We can now confirm that all unsolved cases had extensive testing of all known albinism-related genes (including CNV analysis). The following statement has now been included in page 14 (lines 297-299): “*comprehensive genomic testing (including single-nucleotide variant and copy number variant analysis involving the aforementioned 19 albinism-related genes) was performed in all cases that remained unsolved.*”

Comment #4.6R: *Within the 100K_GP cohort, the proportion of cases that meet the inclusion criteria seems to be nearly double (~30%) compared to Bordeaux cohort (~17%). One would expect within the same ancestry groups that these number should be similar.*

Response #4.6R: One of the inclusion criteria is ancestry and the above statement involves circular reasoning. It is however true that the diagnostic rate of albinism testing was higher in the European subset of the Bordeaux cohort compared to the European subset of the Genomics England cohort (73% vs 59%). On the one hand, one would expect these numbers to be similar. On the other hand though this observation is, to a degree, unsurprising given the variability noted in different cohorts described in the biomedical literature:

- Chan *et al.*, *Genes* 2023 [DOI: 10.3390/genes14010135]: 70% (US; panel-based testing)
- Jackson *et al.* *Am J Med Genet C Semin Med Genet* 2020 [DOI: 10.1002/ajmg.c.31837]: 56% (UK; whole genome sequencing)

- Chan *et al.*, *Genes* 2021 [DOI: 10.3390/genes12040508]: 42% (UK; whole genome sequencing or panel-based testing)
- Mauri *et al.* *J Hum Genet* 2017 [DOI: 10.1038/jhg.2016.123]: 74% (Italy; panel-based testing)

One could argue that the high diagnostic rate in the Bordeaux cohort (73%) supports the appropriateness of our case definition criteria. If these were loose (e.g. only presence of nystagmus or foveal hypoplasia), one would not expect to solve so many cases by testing albinism genes.

Comment #4.7R: *Comment related to 1.2 – thank you for including CNV analysis as part of the variant pathogenicity workup. The authors have mentioned that this has strengthened the detected signals. Presumably this has also further decreased the number of unsolved cases. However in the revised manuscript it is unclear how many were excluded as a result of this. Based on revised figure 1, only 1 case seems to be excluded.*

Response #4.7R: In the analysis described in our initial submission, the study design did not involve determining/defining solved and unsolved cases. We have introduced this in the revised manuscript in response to reviewers' comments. In Figure 1, we show that out of 1,015 cases, 173 met the inclusion criteria.

We have now calculated the additional number of solved cases following CNV analysis in the Bordeaux cohort and this was found to be 90 overall, of which 41 were of European ancestries.

Comment #4.8R: *Comment related to 1.5 – it would be helpful to have the numbers within the figure to readily visualise them.*

Response #4.8R: We have now included the number of probands in an amended Fig.2. It is noted that all numerical data are provided in Suppl. Table 2.

Comment #4.9R: 1.6 – *The endophenotypes of visual acuity and central retinal thickness are influenced by many other factors such as refractive error, media opacities, acquired (known and unknown) retinal disorders, genotype (including carriers). It is also well known that carriers of albinism can have slightly reduced visual acuity and increased central retinal thickness. Have the authors excluded possible albinism carriers (including the pathogenic cis-YQ haplotype) and similarly accounted for other causes of reduced vision when constructing this cohort from UK Biobank. These details are missing. Once again numbers with a possible flow chart would be useful. Similarly please include numbers within supplementary table 4.*

Response #4.9R: We thank the reviewer for the helpful comment.

- To reduce the risk of bias introduced from other ophthalmic conditions, we have now excluded all UK Biobank controls that have been assigned an ophthalmology-related ICD-10 code (“Chapter VII Diseases of the eye and adnexa” in data field 41270).
- As genome sequencing data in all UK Biobank participants is not readily available, it is not possible to identify (and exclude) all individuals who carry rare albinism-related variants. It can however be argued that rare albinism-related variants are unlikely to be a significant confounding factor (in the visual acuity and central retinal thickness analyses). Although the reviewer states that it is “*well known that carriers of albinism can have slightly reduced visual acuity and increased central retinal thickness*”, to the best of our knowledge, there are no studies discussing retinal structure and visual performance in individuals who are heterozygous

for pathogenic variants in most of the 19 albinism-related genes. There is a recent report (PMID: 35379600; published in August 2023) discussing observations in 13 individuals who carried heterozygous changes in either *TYR*, *OCA2* or *SLC45A2*. While this identified subtle differences in OCT-measured retinal thickness, it concluded that there were no significant differences in vision between carriers and controls. In any case, to mitigate some of the risk of confounding, we extracted data for all albinism-related, HGMD-listed variants that were found in the UKB genotype array calls (n= 88; Suppl. Data 4) and excluded individuals who were found to carry these rare changes (in a heterozygous or a homozygous state). This led to exclusion of 19,772 UKB participants (see Suppl. Fig.4).

- with regards to the cis-YQ haplotype, we have now added the status of the *TYR*:c.575C>A and *TYR*:c.-301= changes as covariates in our UK Biobank analysis. To achieve this, we have conducted further analyses using a linear regression approach (Suppl. Tables 5 and 6) in addition to the previously described nonparametric hypothesis testing method (Suppl. Table 4).

Relevant changes were made in the Methods section including:

- page 20, line 488-491: *“Aiming to reduce the likelihood of obtaining spurious signals due to the presence/impact of ophthalmic conditions/features not related to albinism, we excluded all UK Biobank volunteers that did not have the ICD-10 term “Albinism” [E70.3] and were assigned an ophthalmology-related ICD-10 code (“Chapter VII Diseases of the eye and adnexa).”*
- page 19/20, line 465-480: *“Aiming to reduce the likelihood of obtaining spurious signals due to the presence of albinism-related variants other than the two studied changes (*TYR*:c.1205G>A and *OCA2*:c.1327G>A), we excluded UK Biobank participants whose genotyping array data (data field 22418) suggested that they carried an HGMD-listed variant in an albinism-related gene (Supplementary Data 4).”*
- page 19, line 480 & 497; page 20, line 519: *“sex and the number of common albinism-associated alleles were included as covariates”*

Furthermore, a flow chart summarising the approach that we used in UKB participants (including modifications made in response to the above comment) can now be found in a new Figure:

Supplementary Figure 2. Flowchart outlining the UK Biobank case-control study and visual acuity and retinal thickness analyses

Aiming to validate the findings of the primary analysis (Fig.1, Supplementary Fig.1a, Supplementary Table 2), we performed additional studies in an independent cohort, the UK Biobank.

To reduce the likelihood of obtaining spurious signals due to population stratification effects or due to the presence of albinism-related variants other than the two studied changes (*TYR*:c.1205G>A and *OCA2*:c.1327G>A), we focused only on individuals: (i) who were projected to have European-like ancestries; (ii) whose genotyping array data suggested that they did not carry an HGMD-listed variant in an albinism-related gene. The following two covariates were used in logistic and linear regression analyses: sex and number of common albinism-associated alleles, *i.e.* DNA sequence alterations in the genomic locations corresponding to *TYR*:c.-301C>T [rs4547091] and *TYR*:c.575C>A (p.Ser192Tyr) [rs1042602]. These two variants have been previously shown to modify the effect of *TYR*:c.1205G>A which is a common missense change that can act as a “hypomorphic” variant.

The relevant results can be found in Fig.3a-c and in Supplementary Tables 4-6.

PCA, principal component analysis; OCT, optical coherence tomography; HGMD-listed variants, variants with a “total” minor allele frequency (MAF) <1% in the Genome Aggregation Database (gnomAD v2.1.1) and a “disease-causing” (DM) label in the Human Gene Mutation Database (HGMD) v2021.2.

The relevant results are shown in an amended Suppl. Table 4 and in two new Suppl. Tables (5 and 6). Relevant numbers have been included in the footnotes of Suppl. Tables 5 and 6:

For the visual acuity analysis, the number of individuals in the A, B, C, and D groups were as follows: 30,098; 26,190; 499; and 460. For the retinal thickness analysis, the number of individuals in the A, B, C, and D groups were as follows: 6,870; 5,976; 112; and 96.

Comment #4.10R: *Having a validation/replication cohort is important, however the case-control*

comparisons from the UK Biobank are extremely small numbers to draw meaningful conclusions and validity for a replication cohort.

Response #4.10R: Please see also our response to comment #3.1R. In brief, obtaining access to the UKB Primary Care records (resource 591) has allowed us to increase the number of overall number of UKB cases from 24 to 39. By taking into account these additional cases and by modifying our UKB analytical approach in line with the reviewer's recommendations (in Comment #4.10R) has led to the following:

- the UKB case-control odds ratio for genotype group D (dual heterozygotes) has increased from >3.11 to >6.9 (Suppl. Table 3)
- the significant differences that were previously noted in the UKB visual acuity and retinal thickness analyses were still observed. For example the Kruskal-Wallis p-value for vision was 2.5×10^{-6} and now is 1.4×10^{-3} . The Kruskal-Wallis p-value for retinal thickness was 6.3×10^{-5} and now is 8.2×10^{-3} .

On a more general point, as mentioned in responses #3.1R and #4.4R it is possible to obtain valid, statistically significant results even in analyses involving small numbers. In these cases, it is crucial to consider confounding factors and we would like to thank the reviewers for their recommendations which have helped us focus more sharply on potential sources bias. We have now performed a multifaceted analysis using a number of approaches in multiple cohorts and there is still a strong signal for OCA2/TYR dual heterozygosity. We are confident that the balanced, probabilistic interpretation of our results is relevant and appropriate, and reflects the established biological interaction between the TYR and OCA2 molecules.

Comment #4.11R: *1.7 – Since the numbers are small, performing segregation analysis of these variants within pedigrees would provide more substantial evidence whether different albinism traits are truly related to these variants.*

Response #4.11R: It has been possible to obtain segregation analysis data (and relevant clinical information) in family members of four dual heterozygotes from the Bordeaux albinism cohort. These are now shown in a new Suppl. Figure (Suppl. Fig.3). It is noted that despite our best efforts, it has not been possible to obtain further genetic/clinical information on individuals from these families. Notably, as we have highlighted in our previous response to comment #1.7 we have decided to conduct large-scale analyses that focused on populations rather than specific families in order to increase the reliability and generalisability of our findings.

Comment #4.12R: *As mentioned previously a loose definition of albinism based on possible presence of one of two traits is not a robust case definition and the authors should aim to highlight which of the phenotypic spectrum is linked to dual heterozygosity.*

Response #4.11R: We would like to re-iterate that the perception that we defined albinism loosely (“based on the possible presence of one of two traits”) is inaccurate (see also our response to comments #4.1R and #4.6R). We thank the reviewer for giving us the opportunity to clarify this.

With regards to the phenotypic spectrum linked to dual heterozygosity, we have included a Table providing clinical information on dual heterozygous individuals (Suppl. Data 3).

We hope that these changes have addressed the reviewers’ comments and that you will now be able to consider our manuscript as suitable for publication in *Nature Communications*.

Thank you for your help and time,

Yours sincerely,

Panos Sergouniotis, David Green, Vincent Michaud, Graeme Black, Benoit Arveiler and co-authors

REVIEWER COMMENTS

Reviewer #3 (Remarks to the Author):

The authors have answered all the questions, slightly expanded the dataset to increase power. They have applied all possible safeguards to ensure against false association. They also discussed in greater detail the hypothesis of a synergistic or epistatic action of the two loci, with the necessary caveats.

Reviewer #4 (Remarks to the Author):

The authors have discussed part of the concerns in the point-by-point response, thank you for your efforts. However there still remain significant concerns, particularly surrounding case definitions and the Bordeaux cohort. This study aims to investigate the role of two variants TYR:c.1205G>A (p.Arg402Gln) [rs1126809] and OCA2:c.1327G>A (p.Val443Ile) [rs74653330] when in dual heterozygous state – can it lead to albinism (or an increased risk of albinism)?

Fundamental to this question is ensuring that the cases of albinism are “well-defined”. If the aim is to determine whether certain variants are disease causing or increasing risk to a certain disease, it is essential to show how the disease/ diagnosis was established. The authors share that all cases from Bordeaux had both dermatological and ophthalmological phenotypes, moreover these phenotypes were independently assessed by two clinicians. Since these phenotypical characteristics are available, it is important that this data is presented within the supplementary materials at an individual case level (include in supplementary data 1). In addition to skin hypopigmentation, I would recommend presenting this data against established criteria (Kruijt et al. Ophthalmology 2018), and should also include visual acuity, therefore readers can independently determine how strong or weak the evidence is for a diagnosis of albinism (based on phenotype alone). This should be presented alongside the genotypic data (within supplementary data 1) for all 1015 probands.

If foveal hypoplasia/infantile nystagmus were used as part of ocular phenotypic criteria for diagnosis please include details of how this was assessed. Similarly, when grading TID and retinal hypopigmentation – how was this achieved? Was imaging performed or just on clinical examination. Also include details of the OCT used and scan protocols to ensure capture of foveal region. The authors use the term prominent foveal hypoplasia, please specify the grade of foveal hypoplasia and how this was assessed (one of the major criteria for establishing albinism diagnosis, Kruijt et al. Ophthalmology 2018). This will provide further assurances to the level of consistent phenotyping expected in establishing these diagnoses.

Partial glimpses of the level of phenotyping has been provided in supplementary data 3 for a small subset. Within this dataset, I am concerned that more than half do not have a visual acuity measure which is an expected baseline standard for any ocular phenotyping for albinism, 27 out of 33 do not have an entry for optic nerve misrouting (whilst not all centers perform this – even in young children this can be a very useful test to support the diagnosis of albinism – Kruijt et al IOVS 2019), approximately half do not have a status for foveal hypoplasia and grading is only available in 3 out of 33. In the point-by-point response the authors suggest alternative systems to the diagnostic criteria by Kruijt et al. and also cite the work by group based at the University of Iowa (Dumitrescu et al. 2021). The work by Dumitrescu et al. 2021 uses a clinical albinism score where visual acuity is an important parameter (together with a number of other parameters including foveal hypoplasia), however with the degree of missing data I fail to see how either system, diagnostic criteria or a gestalt approach could be applied.

Unsurprisingly, the Bordeaux cohort of probands with albinism have cases of FRMD7 mutations (see supp data 1). If the cohort had phenotyping to the expected standards these cases would have been picked up (for instance these patients, have much better vision than albinism, most have normal

fovea, they do not have optic nerve misrouting). It is unclear why one of the FRMD7 cases is considered unsolved. One should not include FRMD7/SLC38A8 mutations within this cohort. Importantly, if these cases are present, it is plausible that variants in other known genes (CSNB, achromatopsia, PAX6, other retinal diseases/dystrophies etc) that could cause foveal hypoplasia or nystagmus which have not been tested could represent some of the unsolved cases. Therefore, attributing disease risk to these alleles (rs1126809 and rs74653330) might be unsupported if these other differentials are not ruled out (both phenotype and genotype).

Thank you for clarifying the number of cases of dual heterozygosity and also identifying potential related individuals and removing them from your analyses. Based on supp data 2, there are 9 cases from UHB with dual heterozygosity and 2 cases from UKB. Among the 9 cases of dual heterozygosity (supp data 3), there is severely limited phenotype available to really classify these cases as albinism. For instance, individual GSG121656, has no iris transillumination, no phenotype data on skin hypopigmentation, no phenotype data on nystagmus, no evidence of retinal hypopigmentation, no evidence of foveal hypoplasia – it is unclear why clinical genetic testing was performed. This is one example, but there are significant number of similar cases where phenotype is severely lacking to classify confidently as albinism in supp data 3.

In relation to the UKB cases, it would be helpful to clarify whether pathogenic variants (from exome data) been checked for these 2 cases prior to classifying them as genotype D. Thank you for excluding other ocular diseases within the UKB analyses as this could bias your results. Similarly, when looking at endophenotypes within UKB, one must exclude carriers. The authors suggest: "It can however be argued that rare albinism-related variants are unlikely to be a significant confounding factor" and "to the best of our knowledge, there are no studies discussing retinal structure and visual performance in individuals who are heterozygous for pathogenic variants in most of the 19 albinism-related genes". The authors also point out that some of carrier reports are recent (August 2023). These are incorrect since there are publications with OCT evidence showing abnormal retinal thickness and foveal hypoplasia in carriers of albinism which is well known and established in the field. This extends to both carriers of ocular albinism showing foveal hypoplasia (Published in 2018: PMID: 28234808) and oculocutaneous albinism (published in 2022: PMID: 35379600). It is surprising that the authors were not aware of this since some of them were co-authors on another publication in 2022 highlighting carrier changes on retinal OCT: (Lejoyeux R, Alonso AS, Lafolie J, Michaud V, Lasseaux E, Vasseur V, Derrien S, Robert MP, Le Mer Y, Tadayoni R, Arveiler B, Mauget-Faÿsse M. Foveal hypoplasia in parents of patients with albinism. *Ophthalmic Genet.* Epub 2022 Sep).

It is unclear why exclusions were restricted to only HGMD variants and only to array-based genotype calls. Exome data is available within UKB, therefore it would be reasonable to exclude individuals with heterozygous pathogenic or predicted high impact variants in albinism related genes.

Panos Sergouniotis FRCOphth, PhD
Manchester Centre for Genomic Medicine
Manchester University NHS Foundation Trust,
Oxford Road, Manchester M13 9WL, UK

email: panagiotis.sergouniotis@manchester.ac.uk

May 21st, 2024

Re: Third revision of NCOMMS-22-51271 (“The co-occurrence of genetic variants in the TYR and OCA2 genes confers susceptibility to albinism”)

A point-by-point response to the reviewer’s comments can be found below.

REVIEWER 3 COMMENTS

Comment #3_1: The authors have answered all the questions, slightly expanded the dataset to increase power. They have applied all possible safeguards to ensure against false association. They also discussed in greater detail the hypothesis of a synergistic or epistatic action of the two loci, with the necessary caveats.

Response #3_1: We would like to thank reviewer 3 for their helpful and insightful comments which have allowed us to improve our manuscript.

REVIEWER 4 COMMENTS

Comment #4_1: The authors have discussed part of the concerns in the point-by-point response, thank you for your efforts. However there still remain significant concerns, particularly surrounding case definitions and the Bordeaux cohort. This study aims to investigate the role of two variants TYR:c.1205G>A (p.Arg402Gln) [rs1126809] and OCA2:c.1327G>A (p.Val443Ile) [rs74653330] when in dual heterozygous state – can it lead to albinism (or an increased risk of albinism)?

Response #4_1: The aim of our study is not to provide a yes or no answer to the question “does dual heterozygosity for TYR:c.1205G>A (p.Arg402Gln) [rs1126809] and OCA2:c.1327G>A (p.Val443Ile) [rs74653330] cause albinism”. As mentioned in our previous response (#4.4R), among other, “we wanted to obtain an estimate that could assist within a Bayesian framework that

combines clinical and genetic data (similar to a concept described in a publication linked to the American College of Medical Genetics and Genomics [ACMG] (PMID: 29300386)).

We have now included the following mention to this (Supplementary Table 10): *“the main aim of this study was to obtain an estimate of the probability of an individual who is heterozygous for the TYR:c.1205G>A and OCA2:c.1327G>A variants receiving a diagnosis of albinism, with a view to incorporating this in a Bayesian framework that combines genetic and phenotypic data.”*

Comment #4_2: *Fundamental to this question is ensuring that the cases of albinism are “well-defined”. If the aim is to determine whether certain variants are disease causing or increasing risk to a certain disease, it is essential to show how the disease/diagnosis was established.*

The authors share that all cases from Bordeaux had both dermatological and ophthalmological phenotypes, moreover these phenotypes were independently assessed by two clinicians. Since these phenotypical characteristics are available, it is important that this data is presented within the supplementary materials at an individual case level (include in supplementary data 1). In addition to skin hypopigmentation, I would recommend presenting this data against established criteria (Kruijt et al. Ophthalmology 2018), and should also include visual acuity, therefore readers can independently determine how strong or weak the evidence is for a diagnosis of albinism (based on phenotype alone). This should be presented alongside the genotypic data (within supplementary data 1) for all 1015 probands.

Response #4_2: We agree with the reviewer about the importance of phenotyping in genetic studies involving large-scale cohorts and we are grateful for the opportunity to provide more information.

Although the phenotypic characteristics of all the patients in the Bordeaux cohort are available, sharing detailed clinical information at an individual case level alongside genotypic data is not permitted by the terms set in the ethical approval underpinning this work. We are however able to present relevant aggregate data. These are shown in the following new Figure:

Supplementary Figure 4. Prevalence of albinism-related phenotypic features in the University Hospital of Bordeaux albinism cohort (compared to the findings of four related studies)

Prevalence of key albinism-related features in the University Hospital of Bordeaux albinism cohort (“Bordeaux”, n=1,015) compared to the findings of four other studies (that reported both genetic and clinical data in albinism cohorts and included at least 50 individuals) by

- Dumitrescu *et al.* 2021 (US; n=58)
- Kessel *et al.* 2021 (Denmark; n=92)
- Kruijt *et al.* 2018 (Netherlands; n=522)
- Mauri *et al.* 2017 (Italy; n=321)

Bar graphs with 95% confidence intervals are shown.

It is noted that the median visual acuity in the Bordeaux and Mauri *et al.* cohorts was 0.70 LogMAR; in the Kruijt *et al.* cohort this was slightly better at 0.50 LogMAR.

It is also highlighted that the diagnostic rate of genetic testing in these cohorts was reported to be:

- present study (Bordeaux): 75.6% (767/1,015)
- Dumitrescu *et al.* 2021: 34.4% (20/58)
- Kessel *et al.* 2021: 83.7% (77/92)
- Kruijt *et al.* 2018: 64.4% (127/197)
- Mauri *et al.* 2017: 73.5% (236/321).

The above figure highlights that our cohort is not an outlier compared to previously published datasets. Notably, an additional point that we briefly eluded to in our previous response (#4.4R) is that the diagnostic rate of genetic testing in our cohort is greater than that of many other cohorts, supporting the appropriateness of our case definition criteria.

We have now included the following in the penultimate paragraph of the Discussion (pages 12-13, lines 208-213):

“Another potential caveat is around the granularity of the phenotypic information used for case definition (Methods; Supplementary Table 10). The availability of imaging/electrophysiological data in the present cohort for instance did not match that of certain smaller-scale studies^{46,47}.

Nonetheless, the prevalence of key albinism-related phenotypic features in the main dataset was comparable to that in other relevant cohorts⁴⁶⁻⁴⁹ (Supplementary Fig.4).

Please see also our responses to comments #4_4, #4_5 and #4_6.

Comment #4_3: *If foveal hypoplasia/infantile nystagmus were used as part of ocular phenotypic criteria for diagnosis please include details of how this was assessed. Similarly, when grading TID and retinal hypopigmentation – how was this achieved? Was imaging performed or just on clinical examination. Also include details of the OCT used and scan protocols to ensure capture of foveal region. The authors use the term prominent foveal hypoplasia, please specify the grade of foveal hypoplasia and how this was assessed (one of the major criteria for establishing albinism diagnosis, Kruijt et al. Ophthalmology 2018). This will provide further assurances to the level of consistent phenotyping expected in establishing these diagnoses.*

Response #4_3: We have now included the following in the Methods section (page 15, lines 250-258): *“key ocular characteristics of albinism (including nystagmus, iris translucency, retinal hypopigmentation and foveal hypoplasia) were evaluated at the time of ophthalmic examination. The minimum ophthalmic dataset included information on nystagmus, iris translucency (none; peripheral or few iris transillumination defects; diffuse iris transillumination defects)⁴⁸ and foveal hypoplasia (none; hypoplasia / aplasia)^{48,49}. In addition to visual inspection for these features, imaging of the central macula using spectral domain or swept source optical coherence tomography (OCT) platforms was performed in a subset of study participants. Attempts to obtain OCT scans though the foveal centre were made at all times. Relevant images were not acquired in young children and in cases with suboptimal co-operation.”*

We note that, to obtain sufficient power for the conducted analyses, we used the largest cohort of individuals with albinism reported to date. This cohort was collected over 20 years (2003-2022) and involved both paediatric and adult cases from multiple referral centres. As a result, there is a degree of variability relating to the specific phenotyping processes that were followed. This is not

uncommon in the field of rare disease in general and albinism more specifically due to a number of recurrent factors including:

- variability in terms of age at recruitment: the granularity of albinism-related phenotypic information that can be obtained is inevitably smaller in a newborn or infant compared to an adult.
- emergence of new grading systems. For example, the grading system for foveal hypoplasia that the reviewer mentions was first described in 2011 and a novel, less subjective, approach was reported earlier this year (Woertz *et al.* 2024; doi: 10.1167/iovs.65.3.3). For iris translucency (TID grading), a transillumination scale was reported in 2018 (Wang *et al.* 2018, doi: 10.1080/13816810.2017.1342134), with the authors of the relevant publication commenting that the “*de facto* gold standard for evaluating iris transillumination is visual inspection at the time of ophthalmologic examination” (i.e. the approach utilised in our study).
- technological advances associated with phenotyping. For example, image acquisition speed of retinal OCT platforms has increased dramatically by several orders of magnitude over the past two decades. As a result, high-speed OCT devices that allow reliable acquisition of retinal volume scans from the foveal region of young people with albinism/nystagmus are now much more widely available than they were 5-10 years ago.

With regards to the term prominent foveal hypoplasia, we have now included the following clarification (page 15, line 262-264): *prominent foveal hypoplasia (corresponding to high-grade foveal hypoplasia on OCT (≥ 2)⁵⁰ and/or to absence of the foveal depression/reflex as determined by stereoscopic view of the macula by ophthalmoscopy or examination of fundus photographs⁵¹).*

Comment #4_4: *Partial glimpses of the level of phenotyping has been provided in supplementary data 3 for a small subset. Within this dataset, I am concerned that more than half do not have a visual acuity measure which is an expected baseline standard for any ocular phenotyping for albinism, 27 out of 33 do not have an entry for optic nerve misrouting (whilst not all centers perform this – even in young children this can be a very useful test to support the diagnosis of albinism – Kruijt *et al* IOVS 2019), approximately half do not have a status for foveal hypoplasia and grading is only available in 3 out of 33. In the point-by-point response the authors suggest alternative systems to the diagnostic criteria by Kruijt *et al.* and also cite the work by group based at the University of Iowa (Dumitrescu *et al.* 2021). The work by Dumitrescu *et al.* 2021 uses a clinical albinism score where visual acuity is an important parameter (together with a number of other parameters including foveal hypoplasia), however with the degree of missing data I fail to see how either system, diagnostic criteria or a gestalt approach could be applied.*

Response #4_4: While we agree that the information previously presented in Supplementary Data 3 were limited, there are misperceptions about what data is available and what information were

taken into consideration. We apologise for the lack of clarity and we thank the reviewer for the opportunity to provide more information.

In Supplementary Data 3, we initiated presentation of a subset of the readily-available clinical information following a comment from reviewer #1 about the dual heterozygotes in our cohort (*"If a small number of individuals, the clinical details should be provided"*). In addition to data on the 9 dual heterozygotes, in Supplementary Data 3, we included some clinical information on 22 solved cases (for which there is no strong reason to doubt the diagnosis of albinism).

As mentioned in response #4_2, according to the terms set in the ethical approval underpinning this genetic study, directly sharing comprehensive phenotypic data at an individual case level is not permitted and, when we sought advice from our Research Governance Officer, the recommendation was (i) to present aggregate clinical data (which we have now done in Supplementary Fig.4) and (ii) to seek further agreement before presenting additional case-level clinical information.

Following the reviewer's request, we have sought permission to present more detailed clinical data on the 31 study participants discussed in Supplementary Data 3. These are shown in an updated Supplementary Data 3 table which includes additional information presented using an approach analogous to that described by Dumitrescu *et al.* 2021.

More broadly, the reviewer reasonably requests reassurance on two key issues:

- that we have not included an excessive number of non-albinism cases in the Bordeaux cohort. The high rate of confirmation obtained from genetic testing and the fact that, on aggregate, the presented cohort had comparable prevalence of albinism features (and similar mean visual acuity) to other cohorts highlights the robustness of our approach (see response #4_2). Otherwise, we agree with the reviewer's comment that visual acuity is the expected baseline standard for any ocular phenotyping and we have clarified in the Methods section (page 15, line 248) that *"Age-appropriate visual acuity measurements were obtained"*
- that the dual heterozygote group does not include cases that are unlikely to have albinism. We have tried to address this concern by presenting additional phenotypic data and by performing additional genetic analyses (please see our responses to comments, #4_5 and #4_6 below).

Comment #4_5: *Unsurprisingly, the Bordeaux cohort of probands with albinism have cases of FRMD7 mutations (see supp data 1). If the cohort had phenotyping to the expected standards these cases would have been picked up (for instance these patients, have much better vision than albinism, most have normal fovea, they do not have optic nerve misrouting). It is unclear why one of the FRMD7 cases is considered unsolved. One should not include FRMD7/SLC38A8 mutations within this cohort. Importantly, if these cases are present, it is plausible that variants in other known*

genes (CSNB, achromatopsia, PAX6, other retinal diseases/dystrophies etc) that could cause foveal hypoplasia or nystagmus which have not been tested could represent some of the unsolved cases. Therefore, attributing disease risk to these alleles (rs1126809 and rs74653330) might be unsupported if these other differentials are not ruled out (both phenotype and genotype).

Response #4_5: As the reviewer highlights, 6 out of 1,015 (0.6%) individuals in the Bordeaux cohort had disease-associated *FRMD7* variants. Five of these cases are hemizygous male and one is a heterozygous female. The female case is considered unsolved as most female *FRMD7* carriers are unaffected, and the observed phenotype in this case is unlike what is expected in a heterozygote for an *FRMD7* mutation.

The reviewer correctly suggests that it is typically possible to distinguish *FRMD7*-related nystagmus from albinism. However, we respectfully disagree with the comment that the issue with the five cases in our cohort was that phenotyping was not to the expected standards. All these five individuals were very young when recruited and it can often be challenging to make a clear distinction in affected infants, especially in certain contexts of background skin/hair/fundus pigmentation.

The reviewer recommends exclusion of the 5+4 cases with *FRMD7*- and *SLC38A8*-related disease. As these are considered solved, they were anyway not included in the main analysis.

With regards to excluding other differential diagnoses (*PAX6*-related oculopathy, CSNB, achromatopsia etc) in the dual heterozygotes, we have now reviewed the phenotypes and we can confirm that clinically these can be ruled out. More importantly, we have also completed further genetic testing in the 9 dual heterozygotes from the Bordeaux cohort. Analysed genes included:

- *PAX6*
- CSNB-related genes: *NYX*, *CACNA1F*, *GRM6*, *GNAT1*, *RHO*, *PDE6B*, *TRPM1*, *GPR179*, *SLC24A1*, *LRIT3*, *GNB3*, *GUCY2D*
- Achromatopsia-related genes: *CNGB3*, *CNGA3*, *PDE6C*, *GNAT2*, *PDE6H*, *ATF6*

We have now added the following in the penultimate paragraph of the Discussion (page 13, lines 214-218): “*To further reduce the likelihood of an alternative explanation/diagnosis in the TYR:c.1205G>A and OCA2:c.1327G>A dual heterozygotes (Supplementary Data 3), additional genetic studies were undertaken with these confirming that the relevant individuals do not carry disease-implicated variants in PAX6 or in genes associated with achromatopsia or congenital stationary night blindness.*”

Comment #4_6: *Thank you for clarifying the number of cases of dual heterozygosity and also identifying potential related individuals and removing them from your analyses. Based on supp*

data 2, there are 9 cases from UHB with dual heterozygosity and 2 cases from UKB. Among the 9 cases of dual heterozygosity (supp data 3), there is severely limited phenotype available to really classify these cases as albinism. For instance, individual GSG121656, has no iris transillumination, no phenotype data on skin hypopigmentation, no phenotype data on nystagmus, no evidence of retinal hypopigmentation, no evidence of foveal hypoplasia – it is unclear why clinical genetic testing was performed. This is one example, but there are significant number of similar cases where phenotype is severely lacking to classify confidently as albinism in supp data 3.

Response #4_6: We thank the reviewer for the opportunity to clarify this and to provide a more complete outline of the available phenotypic information.

As mentioned in our response to comments #4_4 and #4_6, we have now requested additional permissions to share more detailed, case-level clinical data in the 31 cases discussed in Supplementary Data 3. As shown in the updated Supplementary Data 3 table, GSG121656 has fundus hypopigmentation, foveal hypoplasia, nystagmus and reduced vision (6.3/10; 5/10). Furthermore, his hair, skin and fundus were notably less pigmented than his parents'. Overall, the clinical and genetic findings (including the additional results discussed in Response #4_5) support an albinism diagnosis in this individual.

Comment #4_7: *In relation to the UKB cases, it would be helpful to clarify whether pathogenic variants (from exome data) been checked for these 2 cases prior to classifying them as genotype D. Thank you for excluding other ocular diseases within the UKB analyses as this could bias your results. Similarly, when looking at endophenotypes within UKB, one must exclude carriers. The authors suggest: "It can however be argued that rare albinism-related variants are unlikely to be a significant confounding factor" and "to the best of our knowledge, there are no studies discussing retinal structure and visual performance in individuals who are heterozygous for pathogenic variants in most of the 19 albinism-related genes". The authors also point out that some of carrier reports are recent (August 2023). These are incorrect since there are publications with OCT evidence showing abnormal retinal thickness and foveal hypoplasia in carriers of albinism which is well known and established in the field. This extends to both carriers of ocular albinism showing foveal hypoplasia (Published in 2018: PMID: 28234808) and oculocutaneous albinism (published in 2022: PMID: 35379600). It is surprising that the authors were not aware of this since some of them were co-authors on another publication in 2022 highlighting carrier changes on retinal OCT: (Lejoyeux R, Alonso AS, Lafolie J, Michaud V, Lasseaux E, Vasseur V, Derrien S, Robert MP, Le Mer Y, Tadayoni R, Arveiler B, Mauget-Fajÿsse M. Foveal hypoplasia in parents of patients with albinism. Ophthalmic Genet. Epub 2022 Sep).*

It is unclear why exclusions were restricted to only HGMD variants and only to array-based genotype calls. Exome data is available within UKB, therefore it would be reasonable to exclude

individuals with heterozygous pathogenic or predicted high impact variants in albinism related genes.

Response #4_7: At the time of initial submission, we did not have access to the UK Biobank exome sequencing data. We have now been able to obtain and analyse the relevant dataset (UK Biobank data field 23157: Final exome release). This has allowed us to expand the exclusion criteria in our UK Biobank analysis so that they are in line with the reviewer's recommendation in the above comment. In the Methods section (page 21, lines 434-438), we have now included the following text: "*Aiming to reduce the likelihood of obtaining spurious signals due to the presence of albinism-related variants other than the two studied changes (TYR:c.1205G>A and OCA2:c.1327G>A), we excluded UK Biobank participants whose exome sequencing data (data field 23157) suggested that they carried at least one HGMD-listed variant in an albinism-related gene.*"

Excluding all individuals who carried an albinism-related variant (not only in *TYR* and *OCA2* but also in any of the 19 genes associated with albinism, independently of zygosity or phase) has unsurprisingly reduced the number of UK Biobank cases that were eligible for this replication study (from 24 to 8 individuals; Supplementary Fig.2). However, this has not led to attenuation of the relevant signals; the corresponding section of the Results (pages 10-11, lines 170-175) has been changed

- from: "*We found that UK Biobank volunteers who were heterozygous for the TYR:c.1205G>A and OCA2:c.1327G>A variant combination had not only a higher chance of receiving a diagnosis of albinism (OR>6.9, p-value=0.0004) but also had, on average, slightly worse visual acuity and marginally thicker central retina (the respective Kruskal-Wallis p-values were 1.4×10^{-3} and 8.2×10^{-4} ; Fig.3 and Supplementary Tables 3-6).*"
- to: "*We found that UK Biobank volunteers who were heterozygous for the TYR:c.1205G>A and OCA2:c.1327G>A variant combination had not only a higher chance of receiving a diagnosis of albinism (OR>4.2, p-value=0.005) but also had, on average, slightly worse visual acuity and marginally thicker central retina (the respective Kruskal-Wallis p-values were 0.002 and 0.004; Fig.3 and Supplementary Tables 3-6).*"

Finally, we confirm that inspection of the exome sequencing data from the 8 remaining UK Biobank cases did not reveal any disease-related variants (other than *TYR:c.1205G>A* and *OCA2:c.1327G>A*) in genes linked to albinism, aniridia, achromatopsia or CSNB.

Many thanks

Yours sincerely,

Panos Sergouniotis, David Green, Vincent Michaud, Graeme Black, Benoit Arveiler and co-authors

REVIEWERS' COMMENTS

Reviewer #5 (Remarks to the Author):

The authors propose a significance to the co-occurrence of variants in TYR and OCA2 based on data from the Genomics England 100,000 Genomes Project alongside a cohort of 1,120 individuals with albinism, and investigated the effect of dual heterozygosity for the combination of two established albinism-related variants: TYR:c.1205G>A (p.Arg402Gln) [rs1126809] and OCA2:c.1327G>A (p.Val443Ile) [rs74653330]. These do not propose causality.

the response to the reviewers is satisfactory

this is a innovative thought provoking ms, that we lead to further analysis of all these unclear cases.

Panos Sergouniotis FRCOphth, PhD
Manchester Centre for Genomic Medicine
Manchester University NHS Foundation Trust,
Oxford Road, Manchester M13 9WL, UK

email: panagiotis.sergouniotis@manchester.ac.uk

September 5th, 2024

Re: Nature Communications manuscript NCOMMS-22-51271C (Response to Reviewer)

Please find below a point-by-point response to the reviewer comment

REVIEWER 5 COMMENT

Comment: The authors propose a significance to the co-occurrence of variants in TYR and OCA2 based on data from the Genomics England 100,000 Genomes Project alongside a cohort of 1,120 individuals with albinism, and investigated the effect of dual heterozygosity for the combination of two established albinism-related variants: TYR:c.1205G>A (p.Arg402Gln) [rs1126809] and OCA2:c.1327G>A (p.Val443Ile) [rs74653330]. These do not propose causality. the response to the reviewers is satisfactory this is a innovative thought provoking ms, that we lead to further analysis of all these unclear cases.

Response: We would like to thank reviewer 5 for the feedback and for taking the time to appraise our manuscript.

Kind regards

Yours sincerely,

Panos Sergouniotis, David Green, Vincent Michaud, Graeme Black, Benoit Arveiler and co-authors